# New Insights on the Interaction of Phenanthroline Based Ligands and Metal Complexes and Polyoxometalates with Duplex DNA and G-Quadruplexes

**DOI:** 10.3390/molecules26164737

**Published:** 2021-08-05

**Authors:** Ángel Sánchez-González, Nuno A. G. Bandeira, Iker Ortiz de Luzuriaga, Frederico F. Martins, Sawssen Elleuchi, Khaled Jarraya, Jose Lanuza, Xabier Lopez, Maria José Calhorda, Adrià Gil

**Affiliations:** 1Faculdade de Ciências, BioISI-Biosystems and Integrative Sciences Institute, Universidade de Lisboa, Campo Grande, 1749-016 Lisboa, Portugal; asgonzalez@fc.ul.pt (Á.S.-G.); nabandeira@fc.ul.pt (N.A.G.B.); fdmartins@fc.ul.pt (F.F.M.); mjc@fc.ul.pt (M.J.C.); 2CIC nanoGUNE BRTA, Tolosa Hiribidea 76, Euskadi, 20018 Donostia-San Sebastián, Spain; i.ortiz@nanogune.eu; 3Polimero eta Material Aurreratuak, Fisika, Kimika eta Teknologia Saila, Kimika Fakultatea, Euskal Herriko Unibertsitatea (UPV/EHU), Paseo Manuel de Lardizabal 3, 20018 Donostia-San Sebastián, Spain; jose.lanuza@dipc.org (J.L.); xabier.lopez@ehu.eus (X.L.); 4Laboratoire de Chimie Inorganique, LR17ES07, Faculté de Sciences de Sfax, Université de Sfax, Sfax 3000, Tunisia; euchisawsen3@gmail.com (S.E.); khaledjarraya@yahoo.fr (K.J.); 5Donostia International Physics Center (DIPC), Paseo Manuel de Lardizabal 4, 20018 Donostia-San Sebastián, Spain

**Keywords:** artificial phosphoesterases, DNA, G-quadruplexes, intercalation, LS-DFT, Mo-oxo species, phenanthroline derivatives, phosphoester hydrolysis catalysis, polyoxometalates, weak interactions

## Abstract

This work provides new insights from our team regarding advances in targeting canonical and non-canonical nucleic acid structures. This modality of medical treatment is used as a form of molecular medicine specifically against the growth of cancer cells. Nevertheless, because of increasing concerns about bacterial antibiotic resistance, this medical strategy is also being explored in this field. Up to three strategies for the use of DNA as target have been studied in our research lines during the last few years: (1) the intercalation of phenanthroline derivatives with duplex DNA; (2) the interaction of metal complexes containing phenanthroline with G-quadruplexes; and (3) the activity of Mo polyoxometalates and other Mo-oxo species as artificial phosphoesterases to catalyze the hydrolysis of phosphoester bonds in DNA. We demonstrate some promising computational results concerning the favorable interaction of these small molecules with DNA that could correspond to cytotoxic effects against tumoral cells and microorganisms. Therefore, our results open the door for the pharmaceutical and medical applications of the compounds we propose.

## 1. Introduction

Biomedicine may be considered the cornerstone of modern health care. [1] It includes key enabled technologies such as molecular biology, biotechnology, nanobiotechnology, biological engineering, etc., and concerns a wide range of scientific and technological approaches that range from the understanding of molecular interactions to the study of gene therapy. According to the most recent statistics in Europe, there were 1.93 million deaths caused by cancer in Europe during 2018 (~36.3% of the total number of deaths), which means that cancer is still an important topic for biomedical research and that improvements in chemotherapy are still necessary. On the other hand, the emerging problem of bacterial antibiotic resistance (BAR) has caused 33,000 deaths per year in the EU (https://ec.europa.eu/health/antimicrobial-resistance/eu-action-on-antimicrobial-resistance_en accessed on 1 August 2021) and has become a current hot topic of research. Thus, research developed within these topics for the next years becomes important, timely, and within the priorities of the world concerning health, healthy ageing, and wellbeing. Cisplatin is the reference drug in chemotherapy treatments against cancer. Ref. [2] However, this drug may cause serious side effects due to the damage of normal tissues [3]. Research aimed at the development of more efficient alternatives is valuable. 

The use of flat ligands such as 1,10-phenanthroline (phen), both in their isolated form or included in metal complexes, was devised some years ago as an alternative to cisplatin for chemotherapy treatments [4,5,6]. Their use as an innovative method to fight against BAR has been considered more recently. Refs. [7,8,9,10] They intercalate between DNA base pairs (bps) and can either inhibit the replication of DNA or cleave the DNA chain causing the death of cancer cells or bacteria. Nevertheless, some competition between intercalation and groove binding modes of interaction between these flat ligands and DNA has been proposed in the literature. Refs. [11,12,13,14,15] Indeed, whereas the groove binding occurs very fast (i.e., tenths of a millisecond), the intercalation mode of interaction takes more time to occur (i.e., in the range of few milliseconds). Ref. [12] Moreover, the intercalation mode is usually more related to cytotoxic effects. Since the cytotoxic effect of any intercalator depends on the time of residence of the drug between bps, [16] the design of any efficient drug should aim at an increased drug-DNA interaction to stabilize the intercalated state, but it should do so at less stable groove binding states to make the kinetics faster. Such modulation for the binding sites, and thus for the cytotoxicity, can be achieved by substitution of phen in number and position. For this reason, the comprehension and rationalization of the interactions between flat ligands and duplex DNA (dDNA) becomes crucial. They will give us information about how to modulate the interactions in both modes by substitution in the flat ligands and how to optimize the drug design. In this sense, the effect of substitution in phen in the intercalation process is still not clear. On the other hand, the antibacterial efficiency of phen derivatives is higher when they are coordinated to any metal than when the ligand is alone [10]. However, no satisfactory explanation has been given for it. Thus, the influence of ancillary ligands in the intercalation process is an interesting topic for study. Our contribution to state-of-the-art studies has been the analysis of the nature of the interaction and the investigation concerning how the substitution of flat ligands, such as phen, in number and position (with different kind of functional groups as -CH_3_, -OH, -NH_2_, =O, -Ph, -Cl, -COOH, etc.) favors the intercalation or the groove binding competitive mode of interaction. Moreover, the inclusion of metal atoms has been also analyzed.

Another alternative to overcome the problem with cisplatin is based on the stabilization of G-quadruplexes (GQ), which are alternative non-canonical quadruple-stranded helical DNA structures found in guanine rich sequences of DNA. The formation of GQ gives other singularity to the DNA, which may lead to more selective interactions. Moreover, by avoiding the abovementioned side effects, this new target could replace therapies based on cisplatin. Indeed, the formation and stabilization of GQ were shown to decrease the activity of telomerase [17], which is the enzyme responsible for the elongation of telomeres, a phenomenon that prevents cell apoptosis. Since high telomerase activity is involved in 85% of cancers [18], it is recognized as a potential cancer specific target. Thus, inhibition of telomerase becomes a key method for stopping tumoral cell growth. The presence of GQ in promoters also represents a subject of study. Ref. [19] In this case, the presence of stable GQ in oncogene promoters can alter the expression of the gen reducing some key processes in the growth of tumor cells. Ref. [20] Thus, stabilization of GQ may also be an innovative strategy to fight against BAR since the stabilization of GQ in bacteria by small molecules may also inhibit the expression of genes responsible for BAR. Ref. [21] Several organic ligands, either alone or included in metal complexes, were reported in the literature as GQ stabilizers, inhibiting telomerase activity or disrupting the transcriptional activity of some oncogenes. Ref. [22] There are three main sites in the GQ where these stabilizing small molecules may interact: end-stacking, grooves, and loops (with the first being the most common). In order to favor such interactions, small planar molecules may induce end-stacking binding, whereas the interactions with side loops and grooves are enhanced by the presence of side chains, which are positively charged or have affinity for protons that are attached to the planar aromatic cores. These side chains participate in electrostatic interactions with the negatively charged DNA phosphate backbones. Ref. [23] Moreover, the same organic ligands bound to different metal centers may still generate different GQ affinities and interactions.

On the other hand, and trying to devise other strategies, it was only during the early-mid 2000s that systematic studies on the application of nanostructures based on polyoxometalates (POMs) for cancer treatment became a booming domain. Refs. [24,25,26] Recent advances in the use of POMs for medical applications and their antitumor activity have been reviewed by Bijelic et al. [27] In addition, Bijelic et al. and Kortz et al. also reviewed the latest developments on the use of POMs against BAR. Refs. [28,29] According to these reviews, the proposed modes of antitumoral action of POMs involve several possibilities: (1) the activation of cell death pathways; (2) inhibition of angiogenesis; (3) interaction with proteins; or (4) DNA interaction, among other mechanisms. In the case of POMs action against bacteria the proposed mechanisms considers: (a) inhibition of both PBP2a and β-lactamases; (b) targeting P-type ATPases; (c) impairment of the bacterial electron-transport chain (respiratory system); (d) POM-mediated increase of the reactive oxygen species level via oxidation; (e) interaction with important membrane-anchored proteins and enzymes; (f) disruption of the bacterial cytoskeleton dynamics by POM-interactions with cytoskeletal elements; (g) disruption of the bacterial cell wall leading to leakage of intracellular substances; or (h) interaction with cytoplasmic elements of proteins that are anion-sensitive like nucleic acid-binding proteins. In our studies, among all the processes described in the reviews of Bijelic et al. [27,28], we focused on the interaction of POMs with DNA to promote its phosphoester hydrolysis as artificial phosphoesterases. The seminal works found in the bibliography on this topic are the experimental studies of Parac-Vogt et al. based on [Mo_7_O_24_]^6−^ ({Mo_7_}) species, [30,31,32,33] whereas other experimental works of Abrantes et al. [34,35] were based on the MoO_2_X_2_L species. 

Thus, the work of our team during the last few years has aimed at applying the abovementioned approaches arising from biomedicine to the in silico drug design for cancer therapy and BAR with more than ten works published during the last six to seven years. We have analyzed and rationalized the interaction of small molecules and nanostructures involving phen derivatives and molybdenum with several targets of DNA, not only from a structural point of view but also considering their reactivity. Refs. [36,37,38,39,40,41,42,43,44,45,46,47,48] The general goal of this focused review on our team is to address our new insights on the modelling of the interactions and the behavior of some selected systems, namely phen derivatives and oxo-Mo species with several targets based on canonical and non-canonical DNA to improve antitumoral chemotherapy treatments and to fight against antibiotic resistance. 

## 2. Computational Techniques, Methods and Tools Used in Our Studies

We have used different computational approaches to carry out the different calculations for the three topics developed in these studies. In the lines of research regarding the interaction of small molecules based on phen derivatives and Mo complexes, including phen with dDNA and GQ, we used several models along with different levels of calculation. That is, in the case of sandwich models [49], we used the M06-2X/6-31+G(d,p) level of calculation [50,51,52] with Gaussian09 in order to study the intercalation of phen derivatives with dDNA. Ref. [53] The M06-2X functional is recommended to study systems in which weak interactions such as dispersion are important to explain their behavior (as the intercalation of small molecules between bps of DNA). In the case of the ring models [49] including sugar, phosphates, and Na^+^ counterions, and for the study of the groove binding interactions with the d(GTCGAC)_2_ hexamer, we used the semi-empirical Hamiltonian PM6-DH2 [54] including dispersion effects with the MOPAC software. Ref. [55] In the case of the ring model 2 ps semi-empirical MD, simulations with the PM6-DH2 Hamiltonian were also performed with a time step of 1fs. Explicit water molecules were considered by means of the TIP3PBOX solvent model for the simulations by using a rectangular box with edges no closer than 5 Å to any atom of the solute. Finally, in the case of the studies on the interaction of the Mo[(η^3^-C_3_H_5_)Br(CO)_2_(phen)] complex with dDNA and GQ, we took into account the d(AGACGTCT)_2_ octamer for the dDNA, coming from the 1n37 PDB structure, and the GQ coming from the 2jwq PDB structure to study the interaction of the Mo[(η^3^-C_3_H_5_)Br(CO)_2_(phen)] with this non-canonical structure. These structures were studied at the LMKLL/DZDP level, [56,57] which includes van der Waals corrections. The core electrons were substituted by norm-conserving pseudopotentials. Refs. [58,59] Such LS-DFT computations with ~500 atoms for the dDNA and ~1000 atoms for the GQ were carried out with the Spanish Initiative for Electronic Simulations of Thousands of Atoms (SIESTA) method and associated software. Ref. [60] The cut-off radii for the atomic orbitals of each element were obtained for an energy shift [60] of 30 meV. The tolerances used for the optimizations were 10^−5^ eV for the energetics, whereas the tolerance for the forces was 0.02 eV/Å for the dDNA and 0.1 eV/Å for the GQ. In order to gain deeper insight into the interaction between the small molecules and DNA substrates, we performed the Energy Decomposition Analysis (EDA). Refs. [61,62] To carry out the EDA, we mainly used the B3LYP-D3/TZP level of calculation [63,64,65,66] since the B3LYP-D3 functional includes an explicit Grimme’s D3 correction for dispersion and, therefore, an additional Δ*E_disp_* term appears in the EDA for this functional. For this reason, we thought that the discussion was simpler to visualize better the trends of the intercalated systems. It must be said that the M06-2X/TZP and M06-L/TZP levels were also checked for the EDA. In any case, the three functionals led to results comparable to the MP2/6-31G* (0.25) level of theory already used by Řeha et al. [67] which gave results comparable to the benchmark CCSD(T) data. Ref. [68] These EDA were carried out with the ADF software. Refs. [69,70,71] Another way to gain deeper insight into the interaction between the studied small molecules and DNA structures is the topological analysis of the structures. Two kinds of approaches were used, namely the classical QTAIM developed by Bader et al. [72] and the most recent approach developed by Johnson et al. [73] based on the NCI. The latter provides a rich, 3D representation for the non-covalent interactions with surfaces based on the peaks that appear in the reduced density gradient at low values of ρ. Such isosurfaces are mapped according to values of the sign of the second Hessian eigenvalue, (λ_2_), and while negative values (i.e., stabilizing interactions) are depicted in blue and pale green, positive values (i.e., destabilizing interactions) are represented in yellow and red. QTAIM and NCI computations were performed with the AIM2000 [74] and AIMALL [75] software with the wave functions generated at M06-2X/6-31+G(d,p) level with Gaussian09 in all cases with the exception of the interactions of [Pt(en)(phen)]^2+^ derivatives via groove binding with dDNA in which the wave function was obtained at B3LYP/6-31G(d,p) level with Gaussian09. Finally, for some of these systems in which small molecules are interacting with DNA, we studied the polarization/charge transfer for the different modes of interaction with several charge schemes arising from different approaches (Mulliken, [76] APT, [77] Hirshfeld [78] and NPA/NBO [79,80]).

In the study of the activity of Mo-oxo species as promoters and catalysts for the hydrolysis of the phosphoester bond, we carried out the DFT calculations by using ADF [69,70,71] and the Gaussian 16 Revision A.03 software. Ref. [81] In the case of ADF, geometry optimizations were carried out with the BP86 functional, which uses the Vosko-Wilk-Nusair exchange-correlation potential [82] with the generalised gradient approximation exchange correction reported by the Becke (1998) exchange functional [83] and Perdew (1986) correlation correction [84], as well as Grimme dispersion corrections (BP86-D3). Ref. [66] Relativistic effects were treated with the zero order regular approximation (ZORA) Hamiltonian. Refs. [85,86] The frozen core approximation and triple-ζ Slater-type orbitals (STO) were used to describe the valence shells of C and N (2s and 2p). One polarization function was added to C, N, O, and Mo (single-ζ, 3d, 4f). Triple-ζ STOs were used to describe the valence shells of H (1s) augmented with one polarization function (single-ζ, 2s, 2p). Solvent effects were included with the COSMO [87] and standard parameters (Water, ε = 78.39). Analytical frequencies were calculated to characterise the obtained stationary points and calculate the Gibbs free energies (standard state T = 298.15 K, *p* = 1 atm). Transition states were followed after a fractional displacement of the imaginary vibrational mode to both the reactant(s) and product(s). On the other hand, Gaussian calculations were carried out with the B3LYP hybrid functional. Refs. [63,64,65] The LANL2DZ effective core potential with the associated double zeta basis set supplemented with f polarisation functions was used for Mo atoms [88,89,90,91,92] and the 6-31+G(d,p) basis set [52] for the rest of the atoms. Dispersion effects were included by using the third version of Grimme dispersion with the Becke–Johnson damping approach. Ref. [93] Stationary points were characterised by means of frequency calculations, and the intrinsic reaction coordinate was followed in order to obtain the geometries of the reactants and products followed by unconstrained optimisations. Refs. [94,95] For some mechanistically relevant stationary points the energies were refined with the 6-311++G(3df,2p) basis set. Ref. [96] For these calculations with Gaussian 16 Revision A.03 the Polarisable Continuum Model (PCM) was used to take into account the solvent in an implicit way [97,98]. 

## 3. Studies on the Interaction of Small Molecules with Duplex DNA

During the last few years, the use of phen derivatives (as isolated ligands or forming part of metal complexes) in medical applications using dDNA as target has been focused largely on alternative chemotherapy treatments for cancer. Refs. [4,5,6,99,100] Moreover, these kinds of flat ligands showed promising results against BAR. Ref. [8] In consequence, recent efforts in several scientific fields have started to consider the interaction between phen derivatives and DNA. Refs. [36,37,38,39,40,41,42,46,47,101,102,103,104,105,106,107,108,109,110] Thus, our main goal has been to understand, at a fundamental level, the interaction of phen derivatives with dDNA and how the substitution by means of different functional groups can modulate their interaction, efficiency as intercalators, and their cytotoxicity. Small molecules, including phen derivatives and dDNA, can interact in different modes [15] (see Figure 1):(A)Cross-link interactions occur when the molecule reacts with DNA by forming covalent bonds between two nucleotides [111,112] and the small molecule is placed mainly in the major groove.(B)Groove binding interactions are found when the small molecule is placed in the minor groove (mg) or the major groove (MG), and the interaction with dDNA arises from weak interactions. Usually, this interaction is metastable and is a previous step to the intercalation [11,12,13,14,15,47].(C)In the intercalation, the flat molecules are located between two bps. The interaction is mainly ruled by π−π stacking between the aromatic moieties of the intercalator and the π-system of the bps. Refs. [36,37,38,39,40,41,42,46,67,113] In the case of phen derivatives, substitution by different functional groups may increase or decrease the strength of the intercalation between bps. X-H··Y, CH/π, CH/n, H···H bonds or other weak interactions, not only with the bps but also with the sugar and phosphate backbone, may stabilize intercalation of phen derivatives (as we recently demonstrated). Refs. [39,40,41,42,46] Moreover, intercalation may be achieved in two different orientations, namely through the MG and via mg. (see Scheme 1).

Depending on the number and position of phen functionalization, one orientation or the other may be favored. In addition, for octahedral metal complexes, the weak interactions may also involve the ancillary ligands [40,41].

(D)The insertion mode of interaction between flat ligands and dDNA occurs when such flat molecules are inserted between two bps of DNA that are not matched. Refs. [101,102] In this case, the interaction is preferred from the mg orientation and results in the ejection of mismatched DNA bases with the flat ligand acting as π-stacking replacement.

As we explained above, a competition between intercalation and groove binding takes place when some flat molecule interacts with dDNA. Refs. [11,12,13,14,15] Nevertheless, the inclusion of functional groups in the flat ligand, in different numbers and positions, influences this competition and may be used as a strategy to modulate the cytotoxic behavior, as reported in the work of Brodie et al. [100] dealing with Pt metal complexes including phen and different methylated derivatives (see Figure 2). Brodie et al. [100] observed experimentally that 5-Mephen and 5,6-Me_2_phen reached IC_50_ values of 1.5 ± 0.3 μM and 2.8 ± 0.8 μM against Murine Leukaemia L1210 cell lines. These methylated phen derivatives were more cytotoxic than the rest of methylated phen derivatives, with IC_50_ values of 9.7 ± 0.4 μM for phen and >50 μM for 4-phen, 4,7-Me_2_phen, and 3,4,7,8-Me_4_phen, and the value corresponding to the 5,6-Me_2_phen being close to that of the cisplatin reference (0.5 μM). We proposed [39] an explanation for these experimental behavior based on the EDA, [61,62] and QTAIM [72] and NCI analyses [73].

In our previous studies [36,37,38,39,40,41,42,46,47], we tried to gain insight into the interaction and the eventual cytotoxicity of phen derivatives by means of quantum mechanics (QM) approaches. We proposed that the quantification of the intrinsic contributions to the interaction energy (Pauli repulsion (Δ*E_Pauli_*), dispersion forces (Δ*E_disp_*), electrostatic contributions (Δ*E_elstat_*), and charge transfer and polarization terms (Δ*E_orb_*)) when the flat ligand (intercalator) is functionalized with different groups would provide a good strategy for drug design. However, the solvation effects (for the DNA structure and the intercalator) must be considered and the influence of the ancillary ligands must be analyzed in order to understand the interaction of phen derivatives with DNA. Refs. [38,39,40,41,42,46,47,114] Moreover, the use of QTAIM and NCI analyses provides a very detailed view of the weak interactions between atoms that rule the cytotoxic effects of the small molecules when interacting with DNA.

### 3.1. Current Methods for Modelling the Interaction of Small Molecules with dDNA

There are several ways to address the scientific challenge of understanding the interaction of small molecules, such as phen derivatives, with dDNA. The seminal QM works on the intercalation of flat ligands between bps of DNA were published ~20 years ago. Refs. [67,113] They used three-body models, which were simple models consisting of the intercalator and two bases of DNA (one base pair), without taking into account the effect of sugars and phosphates (see Figure 3).

Conventional MP2 calculations were carried out on these reduced models with double-ζ basis set with polarization and diffuse functions, which were mandatory to reproduce the effects of dispersion in the π−π stacking. Refs. Řeha et al. [67] and [113] also modified the original polarization functions with more diffuse polarization functions to reproduce (more correctly) the effects of dispersion. These three-body models were used during the following years in the works of the intercalation of ligands between bps of DNA employing the QM treatment [115,116,117,118], and have still been used in quite recent works. Ref. [109] On the other hand, other more sophisticated approximations were used, where the model contained chains of DNA (from octamers to dodecamers). Refs. [11,12,14,107,108,119] In this case, the treatment of the system was carried out by means of molecular mechanics (MM), classical molecular dynamics (MD), and/or the hybrid QM/MM approximations. Finally, the remaining models reported consisted of the intercalator and two bps, either without taking into account the sugar and phosphate backbone, which corresponds to the sandwich model (Figure 4) [49], or considering it in the ring model (Figure 5). Ref. [49] The computational treatment for these models included semi-empirical methods, DFT with dispersion corrections, MP2, and QM/MM methods [110,120,121,122,123].

We started our research with QM methods applied to the sandwich model interacting with phen derivatives, whereas in the subsequent works the ring models were studied. We considered the non-substituted original phen ligand along with several derivatives including different substitutions. First the hydrogen atoms of the positions four and seven (see Figure 6) were substituted by -OH, -NH_2_, =O, -CH_3_, -Ph, -Cl, and -COOH groups. Refs. [36,37,38,39,40,41,42,46,47] In addition, for =O and -CH_3_ we also analyzed the intercalation when the substitution occurred in positions five and six; for -CH_3_, we also studied monosubstitution in positions four and five, and tetrasubstitution in positions three, four, seven, and eight. Later, for the study of the interaction of Mo phen complexes with dDNA, we took advantage of the SIESTA method and software [60] to go one step beyond the state-of-the-art. We carried out for the first time linear-scaling DFT (LS-DFT) computations in an octamer of dDNA and two isomers of the [Mo(η^3^C_3_H_5_)Br(CO)_2_phen] complex, namely axial (Ax, with phen nitrogen atoms trans to one CO and the allyl, top of Figure 7) and equatorial (Eq, with phen trans to both CO ligands, bottom of Figure 7). As stated above, we chose the LMKLL functional including dispersion effects [56] and the DZDP numerical basis sets [57], where core electrons were substituted by norm-conserving pseudopotentials [58,59].

In the study of the interactions between the phen ligand derivatives and Mo phen complexes with dDNA, we mainly focused on the EDA, the effect of solvent using the continuum model COSMO [87], the analysis of the weak interactions with the QTAIM topologies [72] and NCI plots [73], and the analysis of the frontier orbitals and charges. Our recent results [36,37,38,39,40,41,42,46] show that we need at least a ring model for the correct QM description of any system intercalating phen derivatives in a non-dynamic approach. Nevertheless, if the evolution of these systems with time is studied in a QM approach (semi-empirical in our case) and, considering explicit solvent effects in a box of water molecules (also treated at semi-empirical level in a QM approach), the ring model evolves to a very distorted structure particularly due to the lack of the stacking stabilization provided by the other bps. For this reason, a correct dynamical treatment of the system requires at least a DNA model including not only the sugar and phosphate backbone (as in the ring model) but also one more step above and below the ring model to form a DNA tetramer including the sugar and phosphate backbone. In order to take into account the role of substituents, their important weak interactions with the sugar and phosphate backbone must be also taken into consideration. Thus, the old three-body model and the sandwich model are not able to reproduce properly all the weak interactions, whereas for a MD treatment, a larger model is needed to take into account the rest of the π⋯π stacking and to avoid the distortion of the system during the simulation. These models must therefore be improved by increasing the size of the DNA model for any QM computation, despite the higher computational requirements of the QM treatment compared to MM approaches.

We also performed computations [47] by using the same computational tools, namely semi-empirical methods including dispersion, EDAs, and NCI to study the groove binding interactions of Pt coordination complexes showed in Figure 2 and understand their effect against murine leukemia L1210 described in the experimental works conducted by Brodie et al. [100]. It was possible to explain, at a fundamental level, that the interaction via groove binding was ruled mainly by different kinds of hydrogen bond interactions between the ethylenediamine (en) ligand and the sugar and phosphate backbone and to shed light on the competition between groove binding and intercalation, the latter being the interaction mode that showed cytotoxicity. In addition, it was observed that the Pt atom could also interact with the sugar and phosphate backbone.

### 3.2. Frontier Orbital Analysis: HOMO and LUMO

Some of the electronic properties that were studied in the intercalated systems were the frontier orbitals, namely the HOMO and the LUMO. Their shape for the isolated fragments, namely the intercalators 4,7-(NH_2_)_2_phen, phen, 4,7-(CH_3_)_2_phen, 4,7-(OH)_2_phen, 4,7-Ph_2_phen, 4,7-Cl_2_phen, 4,7-O_2_phen and 4,7-(COOH)_2_phen, and bps, Adenine-Thymine/Thymine-Adenine (ATTA) and Guanine-Cytosine/Cytosine-Guanine (GCCG), are shown in Figure 8. The energies of the HOMO and the LUMO of the associations between intercalator fragments and the ATTA and GCCG bps (mg and MG) were analyzed. Ref. [40] In all cases the energy of the LUMOs of the intercalator fragments was more negative than any LUMO energy of the ATTA or GCCG bps. Therefore, the intercalators act as electron acceptors in terms of polarization effects in the intercalation process. The most negative LUMO corresponds to the 4,7-O_2_phen intercalator and ranges from −4.12 to −4.20 eV, depending on the considered interacting system. It was the strongest π-electron acceptor ligand, while the weakest π-electron acceptor, with the least negative values for the LUMO (−0.62 to −0.70 eV), was the 4,7-(NH_2_)_2_phen ligand.

On the other hand, analysis of the HOMOs indicated that the HOMO of the GCCG bps fragments was less negative than the HOMO of any of the intercalators. Thus, the GCCG bps act as electron donors during the process of intercalation. More subtle is the interaction with ATTA fragments with HOMO energies lower than those of GCCG bps. For all the studied systems, with the exception of those containing the 4,7-(NH_2_)_2_phen, the HOMO energies of the ATTA fragments are less negative than the HOMO energies of the ligands, while those of phen are very close. Therefore, for phen, but especially for the interaction of the 4,7-(NH_2_)_2_phen ligand with the ATTA fragment, the role of each fragment, intercalator or bps, as electron acceptor or electron donor, could change depending on the orientation and final geometries.

### 3.3. Analysis of the Interaction by Means of EDAs, QTAIM and NCI

As commented before, we analyzed the interaction energy between the phen derivatives and the fragments corresponding to the DNA model of the intercalated systems with the EDA. Refs. [61,62] In the EDA, the interaction energy (Δ*E_int_*) between the two interacting fragments is decomposed into different contributions:ΔE_int_ = Δ*E_elstat_* + Δ*E_Pauli_* + Δ*E_orb_* (+ Δ*E_disp_*), (1)

In this equation, the electrostatic term, Δ*E_elstat_*, corresponds to the classical electrostatic interaction between the unperturbed charge distributions of the rigid fragments, Δ*E_Pauli_* is associated with the destabilizing interactions between occupied orbitals, and the orbital interaction contribution, Δ*E_orb_*, comprises the charge transfer and polarization contributions. In addition, if an explicit correction term for dispersion interaction is used, the dispersion correction arises as an extra term, Δ*E_disp_*. On the other hand, if dispersion contribution is part of the functional, then the Pauli repulsion term in the EDA is reduced. Ref. [62] The EDA was carried out with the ADF software. Refs. [69,70,71] Several functionals produced comparable results [36,37,38] and only the results obtained at the B3LYP-D3/TZP level of theory were discussed. In addition, solvent effects were included by means of the COSMO approach [87].

#### 3.3.1. Studies of Phen and Its Keto Derivatives by Using a Sandwich Model

First of all, we analyzed the sandwich model for ketone derivatives. Geometrical arrangements of such intercalated systems are shown in Figure 9.

The results of the EDA performed for these structures are depicted in Figure 10. We considered the intercalation in ATTA and GCCG bps through different orientations (mg and MG). The most important trend observed for these systems was that, although the Δ*E_disp_* contribution was the most important attractive term, it was not enough to cancel the Pauli repulsion contribution. It was also observed that the introduction of O atoms in the phen ligand did not significantly change the attractive Δ*E_disp_* term. Nevertheless, these keto systems displayed more negative values of Δ*E_orb_* than the phen counterparts. Finally, the Δ*E_elstat_* contributions behaved erratically and a general tendency could not be established. The final trends were similar to those for the phen systems. That is, the repulsive Δ*E_Pauli_* term was the most important one. The Δ*E_disp_* contribution was the most important attractive force, but only with the addition of the Δ*E_orb_* contribution and specially the Δ*E_elstat_* term were we able to obtain negative values for the Δ*E_int_* energy.

#### 3.3.2. Studies of Phen and Methylated Derivatives by Using Both Sandwich and Ring Models

We continued studying the intercalation of methylated derivatives and performed geometrical optimizations of different methylated derivatives between ATTA and GCCG bps by taking into account both orientations, mg and MG (see Figure 11). The EDA was also performed for these methylated systems that were also studied by Brodie et al. [100], namely 4-Mephen, 5-Mephen, 4,7-Me_2_phen, 5,6-Me_2_phen, and 3,4,7,8-Me_4_phen ligands. Sandwich models were considered to study their intercalation between ATTA bps. The trends of the EDA are shown in Figure 12.

A striking observation is the increase of the Δ*E_disp_* with the number of -CH_3_ groups. It is not surprising, since the dispersion terms increase with the polarizability of the system that is related with the number of -CH_3_ groups. A similar trend is also observed for the repulsive Δ*E_Pauli_* contribution, but the amounts differ greatly, depending on the orientation of the intercalation. For example, Δ*E_Pauli_* is 63.5 kcal mol^−1^ for the intercalation of the 3,4,7,8-Me_4_phen system through the mg, whereas its intercalation via MG reaches 74.3 kcal mol^−1^. Δ*E_orb_* are more similar for the different systems and vary only from −12 to −17 kcal mol^−1^. On the other hand, Δ*E_elstat_* differs from one structure to another and no trend is detected. At this point, it is important to recall that the most important attractive contributions to the Δ*E_int_* are the dispersion forces, Δ*E_disp_*, but they cannot balance the repulsive Pauli term, Δ*E_Pauli_*, on their own. Δ*E_orb_* contributions and specially Δ*E_elstat_* forces are needed to obtain negative values for Δ*E_int_*. The interaction energy, Δ*E_int_*, increases with the number of -CH_3_ groups for the intercalation of ATTA through the MG but not for the intercalation of ATTA via mg. For example, the 3,4,7,8-Me_4_phen tetrasubstituted ligand intercalates in ATTA through the mg with Δ*E_int_* of −35.0 kcal mol^−1^, less negative than for the disubstituted systems 4,7-Me_2_phen (−36.5 kcal mol^−1^) and 5,6-Me_2_phen (−37.3 kcal mol^−1^).

In order to describe, at a fundamental level, the interactions between the methyl groups and the bps, we performed a topological analysis of the electron density by using the QTAIM methodology. Figure 13 shows some examples from all the studied methylated systems. The 3,4,7,8-Me_4_phen ligand, which may establish more CH/π interactions, binds more strongly than the other methylated systems, when it intercalates between ATTA bps through the MG. Nevertheless, when it intercalates via mg, the number of CH/π interactions is reduced and the disubstituted ligands 4,7-Me_2_phen and 5,6-Me_2_phen display more CH/π interactions than the 3,4,7,8-Me_4_phen ligand (see Figure 13). Therefore, we concluded that, more than the number of -CH_3_ groups, it is the number of effective weak interactions (in this case CH/π interactions) that controls the achievement of a more negative attractive Δ*E_int_*.

Due to the absence of the sugar and phosphate backbone in the sandwich models, the geometrical arrangements are ruled by the weak interactions formed and the geometries obtained do not represent a realistic model of intercalation between DNA bps. For this reason, in our subsequent works at QM level the ring models were considered to highlight the effect of the sugar and phosphate backbone on the weak interactions.

Thus, we performed geometrical optimizations of the intercalation non-substituted phen and its methyl derivatives via mg and MG between GCCG at the PM6-DH2 level (see Figure 14 and Figure 15, respectively).

The EDA was also performed for all the optimized intercalated systems (see Figure 16). It was observed that for all the studied systems, the Δ*E_Pauli_* increased considerably in the ring model. On the other hand, the Δ*E_disp_* still provided the main stabilizing contribution to the intercalation. Nevertheless, Δ*E_disp_* needs the addition of the smaller Δ*E_orb_*, and especially Δ*E_elstat_* forces, to balance the repulsive effect of the Δ*E_Pauli_* forces, which is again in agreement with our previous results for phen and its keto derivatives (Figure 10). This repulsion is significantly large for the 3,4,7,8-Me_4_phen intercalator (106 kcal mol^−1^). We also concluded again, from the EDA studies with the ring models, that the position of the -CH_3_ groups, leading to effective weak interactions, may be more important than the number of groups. Indeed, the 5,6-Me_2_phen ligand intercalated in GCCG bps through mg has the most negative Δ*E_int_* (−33.9 kcal mol^−1^), being thus more negative than that for the 3,4,7,8-Me_4_phen counterpart (−22.5 kcal mol^−1^). This behavior is explained by the higher number of CH/π and CH/n stabilizing weak interactions achieved by 5,6-Me_2_phen, despite the presence of only two methyl groups as will be explained below.

Figure 17 and Figure 18 show the topological analysis of the electron density with the QTAIM and NCI methodologies for some representative studied systems intercalating via mg and through the MG, respectively. It was observed that for the intercalation of the 5,6-Me_2_phen and 5-Mephen ligands via mg, which correlated with the best results of cytotoxicity in the experimental work of Brodie et al. [100] and displayed the most negative Δ*E_int_* in the EDA, the -CH_3_ groups were able to form interactions with O and N heteroatoms of the bps. This is also represented in isosurfaces with a considerable negative value for NCI analysis. These interactions are highlighted with blue arrows (see Figure 17). Moreover, H⋯H interactions were also found with the surrounding sugars, allowing an even better stabilization of the system. Refs. [39,40,41,42,124,125,126,127] On the other hand, the 3,4,7,8-Me_4_phen derivative presented several weak interactions due to the presence of the four Me groups. However, some zones with a high positive value of the second value of the Hessian were also observed in the NCI isosurfaces (highlighted with red arrows), which indicated the increasing steric repulsion with the sugar and phosphate backbone.

In the intercalation via MG, the methyl groups of 5,6-Me_2_phen and 3,4,7,8-Me_4_phen (Figure 18) interacted with the O atoms of the sugars and with the heteroatoms of the bps. H⋯H interactions were also observed in an isosurface with a negative value (blue).

Thus, in our studies aiming at explaining the experimental results of Brodie et al. [100] we concluded that the number of -CH_3_ groups in the phen ligand favored the cytotoxicity, but the position had to be carefully selected to minimize the steric repulsion, the regions close to the sugar and the phosphate backbone being the most unfavorable positions. For this reason, more than the number of substitutions, it was the position of the substitution that had important consequences in the modulation of the cytotoxicity for the methylated phen systems. The solvent effects had also an important role in the process of stabilization of the intercalator between bps, as we shall see afterwards (Section 3.4).

#### 3.3.3. Studies of Phen and Its Hydroxyl and Amino Derivatives by Using of Ring Models

In order to further explore the influence of the functionalization of the phen ligand, we also considered the substitution, at the four and seven positions, with polar groups as -OH and -NH_2_ (4,7-(OH)_2_phen and 4,7-(NH_2_)_2_phen), since they are capable of forming conventional hydrogen bonds with DNA, when intercalating between GCCG bps. First, we observed that the geometry was distorted for some intercalated systems, due to the interactions with the O atoms belonging to sugars (see Figure 19), this distortion being mainly focused on the intercalation via mg.

Interesting trends were observed for all the contributions of the EDA (see Figure 20). The repulsive Δ*E_Pauli_* values were higher for intercalation via mg. This result was attributed to the repulsive interactions between the intercalator and the sugar and phosphate backbone, since the intercalator is closer to it in the intercalation via mg than in the intercalation via the MG. On the other hand, the introduction of -OH and -NH_2_ functional groups in phen had the important effect of increasing Δ*E_elstat_* and Δ*E_orb_* contributions with respect to the unsubstituted phen ligand. Moreover, Δ*E_elstat_* became equal or even more negative than the Δ*E_disp_* forces, owing to the capability of the 4,7-(OH)_2_phen and 4,7-(NH_2_)_2_phen ligands to form conventional hydrogen bonds with DNA. Therefore, substitution of phen with -OH and -NH_2_ significantly changed the nature of the interaction with respect to the keto and methylated derivatives where Δ*E_disp_* was the most important attractive contribution to Δ*E_int_*. It is not surprising considering the dual nature of the forces ruling the hydrogen bonds (i.e., dispersion and electrostatic). In the case of the conventional hydrogen bonds made from hard acids and hard bases, the electrostatic contribution has a major role in the nature of the interactions [128,129], which is reflected in an increase of the Δ*E_elstat_* forces in the EDA. As stated above, another characteristic of the 4,7-(OH)_2_phen and 4,7-(NH_2_)_2_phen ligands is the increase in the value of Δ*E_orb_* compared to the unsubstituted phen ligand. Since Δ*E_orb_* is related to polarization and charge transfer processes, we assigned the trend to the formation of strong conventional hydrogen bonds between the intercalators and DNA. The substitution with -OH and -NH_2_ functional groups was also responsible for the large total Δ*E_int_* resulting from the stronger hydrogen bonds.

The NCI analysis was also performed for these systems (Figure 21) revealing the strong interactions between the polar groups (-OH and -NH_2_) and the O atoms of the sugar and phosphate backbone. Our results indicated that in order to study the intercalation of flat ligands including functional groups, the implementation of the whole system, bps, and the sugar and phosphate backbone was mandatory. 4,7-(OH)_2_phen and 4,7-(NH_2_)_2_phen ligands were able to form conventional hydrogen bonds when intercalated in GCCG through the mg and via MG. This was only possible when using ring models, not only with the bps, but also with the sugar and phosphate backbone. These interactions were stronger than the π⋯π stacking and van der Waals interactions, and thus were more relevant for these two ligands with -OH and -NH_2_ substituents.

The NCI analysis also helped to compare the two considered intercalators, 4,7-(OH)_2_phen and 4,7-(NH_2_)_2_phen, the main difference being that the -OH groups tended to form stronger hydrogen bonds (more negative value for the NCI index) with the O atoms of the sugar and phosphate backbone than the -NH_2_ groups, even though -NH_2_ groups could form two bonds.

### 3.4. Solvent Effects in the Interaction

The solvent effects were studied for the intercalation of phen, 4,7-O_2_phen, 5,6-O_2_phen, 4,7-(NH_2_)_2_phen and 4,7-(OH)_2_phen in ATTA and GCCG bps and for the intercalation of 4-Mephen, 5-Mephen, 4,7-Me_2_phen, 5,6-Me_2_phen and 3,4,7,8-Me_4_phen in GCCG bps by means of a continuum model with the COSMO approach. Ref. [87] We observed in all the analyzed cases that, as a general trend, the Δ*E_Solv_* solvation penalty (defined as Δ*E_Solv_* = *E_Solv_*(total system)−[*E_Solv_*(intercalator) + *E_Solv_*(pocket)]) ranged from 6.1 kcal mol^−1^ for the 4,7-(NH_2_)_2_phen ligand to 26.8 kcal mol^−1^ for the 4,7-(OH)_2_phen ligand the former intercalating in the ATTA bps via mg and the latter intercalating in the GCCG bps through the MG for the sandwich models. In the case of the ring models the Δ*E_Solv_* ranged from 8.6 kcal mol^−1^ for the (GC/phen/CG)mg system to 35.0 kcal mol^−1^ for the (GC/4,7-(OH)_2_phen/CG)mg system. Thus, the Δ*E_Solv_* penalty was more important for the ring models as a general trend. Moreover, when solvent effects were considered, the energy order was switched and the order in Δ*E_aq_* (defined as Δ*E_aq_* = Δ*E_Solv_* + Δ*E_int_*) was different from that of the Δ*E_int_*. Thus, the inclusion of solvent effects drastically changed the stabilization and could reverse its order.

### 3.5. Metal Complexes Including Phen and the Important Effect of the Ancillary Ligands

Since the strongest hydrogen bonds were formed between the intercalator and the sugar and phosphate backbone, it is clear that the ring model is necessary to study complexes with coligands, which may act as acceptors and donors in hydrogen bonds. Thus, the sandwich model and the former three-body models used in the seminal works of Bondarev et al. [113] and Řeha et al. [67] cannot be used any more for the correct representation of the system, even at the QM level. Keeping in mind this idea and the availability of the SIESTA method and software [60] to carry out LS-DFT computations, we analyzed the interaction of Mo(II) complexes (Figure 7) including phen with DNA with an octamer of DNA, which for the first time included 8 bps and the sugar and phosphate backbone.

We mentioned above the cytotoxic activity of the [Mo(η^3^-C_3_H_5_)Br(CO)_2_(phen)] complex against several tumoral cell lines. Ref. [4] This complex had two possible isomers Ax (Figure 7 top) and Eq (Figure 7 bottom). In the Eq isomer, the two N atoms of phen were coordinated trans to the carbonyl ligands, while in the other isomer, Ax, one N atom of phen was coordinated trans to one CO and the other trans to the allyl. They had very close relative energy and were fluxional in solution, but Ax was observed in the solid state. We analyzed for both isomers, Eq and Ax, the intercalation mode of interaction with dDNA by considering two orientations: (1) via the mg (Eq/mg and Ax/mg) and (2) through the MG (Eq/MG and Ax/MG), as shown in Figure 22. In addition, as the Ax systems had more negative formation energies than their Eq counterparts in the intercalation, as observed in Figure 22, we also studied its groove binding mode of interaction.

To explore the conformational space for the interactions of the [Mo(η^3^-C_3_H_5_)Br(CO)_2_(phen)] complex with dDNA, we carried out docking computations through the HEX software. Ref. [130] We obtained hundreds of structures subsequently optimized with different methods taking advantage of the recently developed semi-empirical methods including dispersion [54], which is an important contribution in the study of biological systems [131], and also the LS-DFT method belonging to the SIESTA software [60], with the LMKLL functional, which includes van der Waals corrections [56], and DZDP basis sets [57] with pseudopotentials [58,59].

The most stable structures for the intercalation modes of interaction are shown along with their formation energies in Figure 22. The most stable groove binding structure for the Ax systems, which led to more stable intercalation modes, is also represented. The intercalation was always more stable than the minor groove binding mode, and the intercalation via mg had more negative formation energies for any isomer Eq or Ax than the corresponding intercalation via MG.

We carried out the EDA [61,62] at B3LYP-D3/TZP level with the ADF software [69,70,71] (see Figure 23), the solvent effects were considered with COSMO, [87] and we performed the NCI analysis [73] associated with the QTAIM topologies. Ref. [72] We show the solvent contributions in Table 1 and the NCI plots in Figure 24 for reduced models of the most stable optimized systems of Figure 22.

The most stable intercalation of the [Mo(η^3^-C_3_H_5_)Br(CO)_2_(phen)] metal complex occurred for the Ax isomer through the mg, and the mg binding for the Ax isomer was less favored than any intercalation mode of interaction, mg or MG, in agreement with the more negative values of the Δ*E_int_* from the EDA, while Δ*E_int_* is less negative for the mg binding interaction (Ax isomer) than for all modes of intercalation. Nevertheless, the nature of the interaction was the same for all modes of interaction, the Δ*E_disp_* being the most important attractive force to the interaction with the dDNA. If we define Δ*E_steric_* as the sum Δ*E_Pauli_* + Δ*E_elstat_*, [132] the Δ*E_disp_* is always higher than the Δ*E_steric_* for both isomers, Ax or Eq. In addition, the smaller Δ*E_orb_* contribution, related to the charge transfer and polarization effects, is also attractive and contributes to a more negative Δ*E_int_* interaction energy.

This behavior was explained by the analysis of the electron density. Figure 24 shows the NCI isosurfaces for the different weak interactions between the Ax and Eq isomers of the complex with the dDNA octamer. In the case of the Ax isomer, we analyzed the two kinds of intercalation modes and the groove binding mode of interaction. The importance of the extended π⋯π stacking interactions between the phen ligand of the Ax isomer of the complex and bps in stabilizing the intercalation modes of interaction was demonstrated, since those extended π⋯π interactions did not appear in the mg binding mode of interaction.

Finally, the solvent effects revealed a slightly larger Δ*E_Solv_* energy penalty for the mg groove binding interaction than for any intercalation mode of interaction of the Ax isomer. This higher Δ*E_Solv_* penalty, combined with the less favored Δ*E_int_* and formation energies, explained why the groove binding interaction was thermodynamically less favored than any intercalation mode.

We also demonstrated that the two isomers, Ax and Eq of the [Mo(η^3^-C_3_H_5_)Br(CO)_2_(phen)] metal complex, preferred the intercalation via mg, as both formation energies and interaction energies, Δ*E_int_*, were more negative for this intercalation. This derived from the presence of the ancillary ligands, because the π⋯π stacking associated to the phen ligand was similar for both the mg or MG orientations of the intercalation, as observed in the NCI analysis and, therefore, the difference in the stabilization was ruled by the ancillary ligands, especially the Br and the allyl ligands in the case of the Eq isomer and the CO ligands for the Ax isomer (due to the different ligand distributions). These ancillary ligands added extra stabilizing interactions (~20–25 kcal mol^−1^) to the intercalation of the [Mo(η^3^-C_3_H_5_)Br(CO)_2_(phen)] complex with the bps, compared to the phen ligand alone. Refs. [36,37,38,39,40,41,42] In addition, in the intercalation through the mg, these weak interactions also took place with the sugar and phosphate backbone and they were very different from those in the intercalation via MG. Indeed, for the intercalation of any isomer, Ax or Eq, of the [Mo(η^3^-C_3_H_5_)Br(CO)_2_(phen)] metal complex with dDNA, the intercalation via mg led to a higher number of weak interactions than via MG. As observed in the NCI analysis, most of these weak interactions between the [Mo(η^3^-C_3_H_5_)Br(CO)_2_(phen)] complex and the dDNA arose from the sugar and phosphate backbone in the case of the intercalation through the mg. It shows the important role not only of the ancillary ligands but also of the sugar and phosphate backbone in stabilizing the intercalation via mg by increasing the weak interactions, which, associated with the Δ*E_disp_* in the EDA, balance the Δ*E_steric_* contribution and led to stable systems. Nevertheless, we have still the external contribution of the solvent effect. The Δ*E_Solv_* penalty in our continuum approximation was slightly more important for the intercalation via mg than for the intercalation through the MG, which led to a very similar final energy balance between the intercalation through the mg and via MG. This effect was more pronounced for the Ax isomer of the [Mo(η^3^-C_3_H_5_)Br(CO)_2_(phen)] complex.

Thus, our findings on the importance of the role of the ancillary ligands on the interaction of the [Mo(η^3^-C_3_H_5_)Br(CO)_2_(phen)] complex with dDNA prompted us to propose an additional point to the strategies of drug design usually based on changes in the metal atom and substitution of the phen ligand in number and position with different functional groups. We believe that changes in the rest of the ligands of the [Mo(η^3^-C_3_H_5_)Br(CO)_2_(phen)] complex, trying to modulate their interaction not only with bps, but also with the sugar and phosphate backbone, will improve the efficiency of the interaction and their biological activity.

### 3.6. Studies on the Groove Binding Interaction of Phen Derivatives with dDNA

More recently we focused our research on the studies of the processes that ruled the groove binding interactions. We focused again on the coordination complexes used in the studies of Brodie et al. [100] addressing the [Pt(en)(phen)]^2+^ coordination complex and its methylated derivatives interacting with the d(GTCGAC)_2_ dDNA hexamer. We optimized several systems by modelling the interaction of [Pt(en)(phen)]^2+^ with the d(GTCGAC)_2_ hexamer via groove binding at the PM6-DH2 level. The most stable optimized geometry obtained revealed that both the en group and the Pt atom were involved in the interaction since the whole metal complex was located inside the groove. This was corroborated with the analysis of the electron density. The NCI analysis (see Figure 25) showed several interactions between the H atoms of the en group with the O atoms of the sugar and phosphate backbone reflected in isosurfaces with high negative values for λ_2_. Also, the Pt presented strong interactions with H atoms of the sugars. In the case of the phen ligand, weak interactions were also detected with the dDNA with cyan and green isosurfaces. Nevertheless, the weight of the strength of the interaction was mainly ruled by the Pt atom and specially the en ligand with dark blue lenticular isosurfaces.

## 4. Studies on the Interaction of Small Molecules with DNA G-Quadruplexes

The possibility to consider non-canonical GQ DNA structures as specific targets for cancer and/or BAR has been explored during the last years. Refs. [17,18,19,20,21,22,23] In the case of cancer, it is known that the immortality of the tumoral cells comes from the constant increase in the length of telomeres. Chromosomes, which are long DNA molecules with part of all the genetic material for any organism, are found inside cells. At the end of these chromosomes, there are some regions containing repetitive nucleotide sequences called telomeres. The function of telomeres is to protect terminal regions of chromosomal DNA from progressive degradation. Actually, the shortening of telomeres is associated with apoptosis and cell death and, on the contrary, keeping the length of telomeres assures the life of cells. An enzyme called telomerase has the function of adding repeat sequences to the telomeres in order to keep their length. In fact, cancer cells are characterized by an abnormal overexpression of telomerase activity, which keeps continuously lengthening the telomeres of tumoral cells and making them immortal. On the other hand, since repeated sequences of nucleotides are found in the telomeric region, it would be possible to take advantage of the formation of GQ in rich guanine telomeric regions to stop the uncontrolled growth of tumoral cells. Indeed, it was found that the stabilization of GQ inhibited the activity of telomerase and that this GQ stabilization could be used to stop the growth of tumoral cells and kill them (specifically since telomerase is overexpressed in cancer cells and thus could be used as a specific target strategy). The stabilization of GQ may be also used to modify the expression of oncogenes in a similar mechanism. Moreover, this alteration of expression of oncogenes by means of stabilization of GQ could be used in the case of the BAR to alter the expression of the gene inducing resistance to antibiotics in bacteria [21].

These GQ may be defined as non-canonical DNA structures where four guanine bases form a square planar array or G-tetrad (see Figure 26). The guanine bases of these G-tetrads are held together by means of Hoogsteen hydrogen bonds stabilizing the system. The stacking of the G-tetrads produces the formation of the GQ. In addition, an extra stabilization of the structure is conferred by alkali cations, especially K^+^ which are found between two tetrads interacting with O_6_ atoms of the guanine bases of both G-tetrads and build the ion-channel in the GQ (see Figure 27).

While recently compiling the most important achievements on the computational modelling of GQ during the last 10 years, [43] we found that the computational studies of these non-canonical DNA structures and their interaction with small molecules have been explored in mainly three ways.

The first approach, mainly used by Sponer et al. [133,134,135,136,137] considered classical MD simulations. In general, parmbsc0 was found as the force field that led to the best results. It must be said that the work of MD simulations was used in different studies with the objective to reproduce the conformational variety of the loops in GQ. Nevertheless, due to the lack of parameters and the limited performance of current force fields, more work is needed in this area. Many efforts within the topic on development and reparameterization of force fields have been carried out during the last years. MD was also the usual choice to study the processes of folding and unfolding of the GQ of DNA, where the good description of weak interactions was very important. Thus, the choice of the force field was also critical for a good description of these processes as well as the achievement of long-time simulations. In addition, classical MD simulations have been carried out during the last few years to analyse the stabilization of GQ by means of ions and small molecules such as organic ligands and metal complexes. Such studies, aiming at the comprehension and rationalisation of the stabilization of GQ through the interaction with ions and small molecules, are common in the bibliography.

In this sense, we also have the QM/MM methods, considered to be the state-of-the-art methods for the study of systems where any ligand or metal complex interact with DNA. In this sense, we highlight the studies of Barone et al. Refs. [138,139,140] The two-layer ONIOM approach was used to optimize the interaction of salen and Schiff-base metal complexes with GQ structures. The presence of high-valent metals required a QM treatment due to the limitations of classical force fields in reproducing metal-ligands interactions. QM/MM methods combined with MD simulations were also used to study the interaction of a Schiff-base ligand with three different metal centres (Ni, Cu and Zn) and the 1KFI PDB structure corresponding to the h-Telo GQ. Many other QM/MM studies have been reported in the literature. These kinds of studies involving small molecules are very interesting from the pharmaceutical and medical point of view since their results and conclusions could be very useful to devise new small molecules by substitution of ligands and changes in metal atoms to improve the interactions and stabilization of GQ. In this sense, it is important to consider not only the affinity of the small molecules with GQ but also the selectivity favouring their interaction with GQ vs. dDNA.

Finally, a QM approach through DFT methods may also be used to calculate several properties related to the geometrical and electronic structure of GQ structures and their chemical properties by means of reduced models of the GQ. However, one of the main drawbacks of these conventional DFT methods is that they are computationally demanding and, therefore, full DFT treatment is usually limited to systems with a reduced number of atoms. Thus, systems including GQ usually have to be downsized to be studied at DFT level. In this case, only the guanine bases of the GQ are considered, and the studies mainly focused on the analysis of the interaction between G-tetrads as well as on the interaction of G-tetrads with ions localized in the ion channel. In this area, the works of Fonseca-Guerra et al. [141,142,143,144] must be highlighted. Indeed, they used DFT-D with reduced models of different numbers of G-tetrads to study the role of the ions in the channel, the stability of several kinds of GQ (guanine vs. adenine), and the cooperativity of the weak interactions (hydrogen bonds and stacking). One of the main conclusions obtained with the DFT-D approach was that alkali cations were not mandatory for the stabilization of GQ. However, the interactions of alkali cations inside the ion channel with the DNA bases of the tetrads gave some extra stability to the system, helping to keep the non-canonical secondary DNA structure. Another interesting finding obtained with reduced models and DFT-D calculations was that RNA-GQ could be more stable than the DNA-GQ and that this behaviour was assigned to an extra hydrogen bond involving the 2′-OH of the ribose of the RNA with the phosphate O atoms, which gave higher conformational stability to the structure.

The improvement of force fields and in the use of GPUs and alternative MD methods as metadynamics, to enhance the exploration of potential energy surfaces and the use of coarse-grained methods to reach longer times for MD simulations, are important challenges that must be addressed in the forthcoming years. We may also take into account the use of QM, QM/MM, QM/MD, and/or QM/MM/MD by using LS-DFT methods such as SIESTA [60] where we could achieve longer times of simulations or optimizations of thousands of atoms within a QM approach. Actually, this was our choice in order to tackle the analysis of the interaction of small molecules with GQ. The SIESTA approach [60] takes the advantage of the use of strictly localised numerical atomic orbitals as basis sets, which have to be strictly zero beyond a user-provided distance, r_c_, from the corresponding nucleus. These finite-support basis sets are key for the calculation of the Hamiltonian and overlap matrices in O(N) operations. The use of pseudopontentials for the inner electrons also helps to achieve the linear-scaling and therefore the treatment of thousands of atoms at the DFT level. Besides the standard Rayleigh–Ritz eigenstate method, it allows the use of localized linear combinations of the occupied orbitals, making the computer time and memory scale linearly with the number of atoms.

In this sense, we calibrated different methods from the geometrical point of view including QM/MM, semi-empirical methods with dispersion, and LS-DFT including dispersion by using as structural reference the 2jwq system of the PDB. As the most highlighting results, we showed that the QM/MM approach, the most popular in the bibliography for this kind of biological systems having ~1000 atoms, may lead to wrong results as depicted in Figure 28.

In Figure 28, we see clearly that the structure of the 2jwq system, which is representative of the interaction of a drug with the GQ, is disrupted, with all the bases fallen down after the QM/MM treatment at M11-L/6-31+G(d,p):UFF level. For this reason, we needed alternative approaches for the treatment of this kind of systems and therefore we tried semi-empirical methods and LS-DFT with added dispersion corrections.

Figure 29 shows the superposition of the original structure 2jwq with each of the structures resulting from three different approaches: (1) semi-empirical methods PM6-DH2 (left); (2) PM7 (middle); and (3) a LS-DFT methodology, including dispersion at the LMKLL/DZ2P level of theory with the SIESTA software (right). All these methods reproduce the original 2jwq PDB structure much better than the QM/MM approach at M11-L/6-31+G(d,p):UFF level and in no case was the falling down of the bases observed. Moreover, the root mean square deviation (RMSD) did not vary much (0.24–2.06 Å). We concluded that all of the approaches, including semi-empirical with dispersion and LS-DFT with van der Waals corrections, behaved in a similar way for this kind of systems from a geometrical point of view.

The conclusion that the semi-empirical PM6-DH2 and PM7 and the LMKLL/DZ2P methods could reproduce geometries better than a QM/MM approach, led us to finish the geometrical calibration of the methods and dismiss the most popular QM/MM approach for these systems. Thus, we only considered the semi-empirical alternatives (including dispersion) and the LS-DFT approach (including van der Waals corrections) at the LMKLL/DZ2P level with SIESTA for the energetic calibration.

Indeed, there is also another relevant aspect to take into account, namely the accuracy of the energetics. For this reason, we performed some calibration of the energies by addressing some tetrads interacting with alkali cations studied by Fonseca-Guerra et al. [144]. We calculated the interaction energies considering the recently developed DLPNO-CCSD(T), which is a near LS-CCSD(T) method [145] as the benchmark for computations and reference. We already have experience in the use of conventional CCSD(T) methods on small molecules and biomolecules. Refs. [125,146,147,148,149] The CCSD(T) highly correlated methods with large basis set, at DFT geometries, is expected to provide accurate numbers for determining interaction energies. Actually, after the boom of DFT methods some years ago, several studies had reported the good performance of a G2 modified composite methodology where the MP2 geometries and HF frequencies were substituted by the DFT ones and the QCISD(T) computations were replaced by CCSD(T) ones. Refs. [150,151] Nowadays, with the LS-DFT and near LS-CCSD(T), we are capable of performing some kind of “near LS-G2 modified composite method” not only with LS-DFT geometries, but also near LS-CCSD(T), in systems with hundreds of atoms (and not only with small systems). These highly correlated methods were restricted to molecules with a reduced number of atoms 20 years ago (tens of atoms) due to the huge requirements of computation time, memory, and disk space. Now, with the recent developments in innovative algorithms, software, and hardware, it is possible to tackle systems with thousands of atoms at the CCSD(T) level. Ref. [152] Focusing on the case of GQ, we used DLPNO-CCSD(T)/def2-SVP results as a benchmark reference for interaction energies and compared the DLPNO-CCSD(T)/def2-SVP benchmark reference values on the interaction energies between G-tetrads and alkaline atoms with the PM6-DH2, PM7, and LMKLL/DZ2P along with the BLYP-D3/TZ2P-ZORA results of the original work of Fonseca Guerra et al. [144]. These results are depicted in Table 2. Our results obtained at LMKLL/DZ2P were very similar to the original results of Fonseca-Guerra et al. at BLYP-D3/TZ2P-ZORA and to the benchmark calculations at DLPNO-CCSD(T)/def2-SVP, whereas the results coming from the PM6-DH2 and PM7 are very different from the benchmark. Thus, even though the semi-empirical methods containing dispersion corrections PM6-DH2 and PM7 performed excellently for the geometries of GQ and G-tetrads, they failed when analysing the energetics of the systems giving a considerable error. On the other hand, LS-DFT calculations with SIESTA at the LMKLL/DZ2P level of theory led to results comparable not only with the published results of Fonseca-Guerra et al. [144] but also with the highly correlated benchmark calculations at the DLPNO-CCSD(T)/def2-SVP level.

Thus, in order to proceed to the study of the interaction of the Ax and Eq isomers of the [Mo(η^3^-C_3_H_5_)Br(CO)_2_(phen)] metal complex with the GQ, our procedure was as follows. We retrieved the 2jwq reference structure from the PDB and the original ligand was removed. Subsequently, we carried out a conformational study with docking (Hex) to determine how both isomers Ax and Eq of the complex [Mo(η^3^-C_3_H_5_)Br(CO)_2_(phen)] could interact with the considered GQ. We only used the Hex software for the docking calculations, but we used different protocols to have different starting points. For the docking screening, we only saved the systems in which the [Mo(η^3^-C_3_H_5_)Br(CO)_2_(phen)] complex was interacting with the bases of the GQ, and we obtained 29 clusters with a RMS threshold of 1.5. These structures were optimized at the LMKLL/DZ2P level. The three most stable structures for each isomer, Ax and Eq, of the [Mo(η^3^-C_3_H_5_)Br(CO)_2_(phen)] metal complex interacting with the GQ are depicted in Figure 30.

It is observed in Figure 30 that the LS-DFT method, including van der Waals corrections at LMKLL/DZ2P level, proposes as the most stable system the association containing the Eq isomer of the [Mo(η^3^-C_3_H_5_)Br(CO)_2_(phen)] complex. Here, the Eq isomer lies between DNA bases of the GQ and interacts mainly by the π−π stacking interaction of the phen ligand with the bases of the DNA. In the second most stable position (17.1 kcal mol^−1^) of the Eq isomer interacting within the GQ, there was a similar kind of interaction as before, whereas for the third Eq association, with a relative energy 33.4 kcal mol^−1^ higher than the first system, we found an interaction with the GQ by means of the Br. This last finding may open the door not only to the possibility of modulating this interaction of [Mo(η^3^-C_3_H_5_)Br(CO)_2_(phen)] with the GQ by means of changes in the metal atom and substitutions in number and position in the phen ligand, but also by substituting the halogen atom by another ligand, such as triflate or chloride (as was already done in previous experimental works). Ref. [4] On the other hand, for the Ax systems, the most stable structure was only 2.3 kcal mol^−1^ above the most stable Eq system, and we observed that all the three most stable systems were found inside the non-canonical DNA structure at the end-stacking of the GQ but helping to stabilize the tetrads of adenine, which were, in principle, less stable than G-tetrads. Ref. [141] This would be a pioneer result in these kinds of studies since it would be the first time, as far as we know, that an octahedral metal complex is localized totally inside a non-canonical DNA structure of four strands, not only stabilizing a GQ by end-stacking but, at the same time, stabilizing a non-canonical structure made from adenine bases, which produces less stable A-tetrads by some kind of end-stacking.

We also performed the EDA for the most stable structure of the Ax and Eq isomers of the [Mo(η^3^-C_3_H_5_)Br(CO)_2_(phen)] complex interacting with the GQ. The results are shown in Figure 31.

The results showed interesting trends. First of all, the Δ*E_int_* interaction energy was 13.9 kcal mol^−1^ more negative for the Eq isomer of [Mo(η^3^-C_3_H_5_)Br(CO)_2_(phen)] in the GQ than for the Ax isomer. On the other hand, the nature of the interaction was different from one isomer to the other. In the Eq system, the Δ*E_elstat_* contribution had a value similar to the Δ*E_disp_* contribution, whereas in the case of the Ax system the Δ*E_disp_* was clearly the most important attractive contribution to the interaction. Thus, whereas the interaction of the Ax isomer of [Mo(η^3^-C_3_H_5_)Br(CO)_2_(phen)] was mainly assisted by dispersion forces, in the case of the Eq isomer the electrostatic contribution became more important. Solvation effects were also analysed and the results are depicted in Table 3.

As for the intercalation in the dDNA the inclusion of solvent effects could switch the order of stabilization determined when solvent effects are not considered. Indeed, when solvent effects were considered the most important Δ*E_Solv_* penalty for the Eq isomer (54.0 kcal mol^−1^) reversed the stability order. After considering solvent effects, the most stabilized structure was the system associated to the interaction of the Ax isomer of [Mo(η^3^-C_3_H_5_)Br(CO)_2_(phen)] with the GQ. In this association, the [Mo(η^3^-C_3_H_5_)Br(CO)_2_(phen)] complex was totally inside the non-canonical DNA structure, which is a very interesting result.

Finally, in order to gain insight on the affinity and selectivity of the interaction of small molecules with dDNA vs. GQ, we must say that the [Mo(η^3^-C_3_H_5_)Br(CO)_2_(phen)] complex should have more affinity for the GQ since the Δ*E_int_* and Δ*E_aq_* are more negative than those for the dDNA (as can be checked, by observing and comparing results for the dDNA above in the Section 3.5. and in references [40,41] with the results in this section). Thus, any of the isomers, Eq or Ax, of the [Mo(η^3^-C_3_H_5_)Br(CO)_2_(phen)] metal complex has more selectivity for the GQ DNA structure.

## 5. Studies on the Use of POMs and Mo-Oxo Species as Artificial Phosphoesterases for the Phosphoester Bond Hydrolysis

In this section, we incorporate (in our computational studies) the reactivity of DNA model targets addressing the hydrolysis of the phosphoester bond in the presence of Mo-oxo species. Indeed, during the last years our team has studied POMs (see one example in Figure 32) and simpler oxides as promoters and catalysts of the phosphoester bond hydrolysis (that is, the use of metal-oxo species as artificial phosphoesterases) [44,45].

The phosphoester bond plays a major role in biological systems and is associated with several biomolecules such as DNA, RNA, ATP, ADP, etc. It is obtained by means of a condensation reaction in which one hydroxyl group of the phosphoric acid reacts with one hydroxyl group of other neighboring molecules to produce an ester bond and to release an H_2_O molecule. In the case of nucleic acids, this process occurs when the -OH groups of the phosphoric acid react with the sugars, riboses, or deoxyriboses found in the different nucleic acids (see Scheme 2). Although these kinds of bonds have a favorable hydrolysis from a thermodynamic point of view, they are kinetically very inert and for this reason they play an important role in several biological functions in maintaining the integrity of the genetic code. At this point, it must be said that the half-life for the hydrolysis of the phosphoester bond in DNA at neutral pH and 298 K is estimated to be 130,000 years. Ref. [153] This opposition to hydrolysis arises mostly since the negatively charged phosphate strands repel the nucleophiles that are also usually negatively charged systems. As a consequence, an important Coulombic repulsion must be overcome to allow the hydrolysis reaction. Ref. [34] Thus, there is a challenge in getting a deeper insight in understanding and rationalizing the catalytic mechanism involving POMs and the negatively charged environment of the phosphoester bonds.

The studies on the mechanisms of the hydrolysis of the phosphoester bond are still a hot topic of research and recent works have been found in the literature. Refs. [154,155,156] Depending on the differences among substrates, such as the nature of the leaving group, protonation state, number of phosphates in the substrate, the characteristics of the catalyst and the properties of the medium in which the reaction takes place, etc., the process may occur in different ways. Thus, whereas for phosphoester dianions with excellent leaving groups in alkali media the mechanism goes through a solvent-assisted dissociative transition state, the pathway with poor leaving groups involves a substrate-assisted associative pathway where the phosphorane transition state is stabilized by a proton shuttle from the substrate to the phosphate moiety concerted with the alkoxide rejection (see Figure 33). Refs. [157,158,159,160] In this transformation, a series of proton transfers to activate the nucleophilic attack and the leaving group is required. Refs. [161,162] Moreover, in acidic media, the monoanionic phosphate transfers the proton of the adjacent O atoms to the leaving group and activates the cleavage of the phosphoester bond. Ref. [163] The fragmentation of the phosphoester bond is the most difficult step [164] and different promoters and catalysts may make one mechanism or the other easier.

The seminal works trying to tackle the mechanism of action of POMs were reported by Parac-Vogt et al.[30,31,32,33] who used the {Mo_7_} polyanion and model substrates such as para-nitrophenylphosphate (pNPP) and many different experimental techniques such as NMR, UV-Vis, RAMAN, etc. They studied the hydrolysis of pNPP in the presence of a {Mo_7_} solution and proposed a mechanism based on the abovementioned experimental techniques. Ref. [31] They found that the [Mo_5_O_15_(PO_4_)_2_]^6−^ species was formed (see Figure 34) and, based on their evidence, they proposed two transformative steps leading to the hypothetical intermediate species [(pNPP)_2_Mo_5_O_21_]^4−^ and [(pNPP)_2_Mo_12_O_36_(H_2_O)_6_]^4−^.

We used computational methods to gain insight into this transformation. Our results surprisingly showed that an anion of the type [(pNPP)_2_Mo_5_O_21_]^4–^ was unlikely to be an intermediate species if compared with the non-catalyzed mechanism (see Figure 35 and Figure 36), since the reaction barrier is significantly higher.

An initial experimental ESI-MS survey of the {Mo_7_} + pNPP reaction mixture provided potential intermediate candidates for the activated hydrolysis of the pNPP. It followed the assumption that molecular fragmentation was limited and that most of the structures present in solution were stable under ionization conditions. In light of the experimentally detected species, we proposed an alternative pathway in which the original {Mo_7_} was in dissociative equilibrium with two other structures in solution: (1) a binuclear {Mo_2_} and (2) a pentanuclear {Mo_5_} structure. The binuclear {Mo_2_} structure was found to be the active species activating the hydrolysis while the {Mo_5_} captured the resulting phosphate. Our working hypothesis turned out to be entirely consistent with the mass spectrometry experiments and the computational data.

Two possible mechanisms are traditionally proposed for the non-catalyzed reaction (Figure 35). As stated above, these are the substrate-assisted mechanism, where the phosphate acts as a proton acceptor of the incoming H_2_O molecule, and the solvent-assisted mechanism, in which there is a straightforward nucleophilic substitution of nitro-phenolate with water. The activation barriers we found for the substrate assisted (ΔG^‡^ = +29.8 kcal mol^−1^) and solvent assisted (ΔG^‡^ = +21.4 kcal mol^−1^) mechanisms were in agreement with previous computational results. Ref. [161] Since the product detected in the work of Parac-Vogt et al. [31] was [Mo_5_P_2_O_23_]^6–^, we should be able to reproduce this phosphate templating reaction by means of the hydrolysis of grafted pNPP to obtain this product. However, the formation of the pentacoordinate phosphorus intermediate had an activation barrier of 44.1 kcal mol^−1^, which was a considerable increase regarding the non-catalyzed reaction (29.8 and 21.4 kcal mol^−1^ for substrate and solvent assisted mechanisms, respectively, compare Figure 35 and Figure 36).

The only adduct with pNPP that was observed in the ESI-MS studies was the dinuclear entity ([Mo_2_O_8_(NO_2_C_6_H_4_PO_4_)(Na)_5_(H_2_O)H]^–^). On account of this observation, we decided to explore the possibility of some mechanism in which a dinuclear species such as [Mo_2_O_4_(OH)_4_(pNPP)]^2–^ could activate the substitution of ligands in the organic phosphate group (that is, two edge-sharing octahedral units bridged by two hydroxyl groups). Several proton isomers were pre-screened and we took into account the most stable species. The results of the hydrolysed activation with {Mo_2_} are presented in Table 4.

Compared to the values for the non-catalyzed mechanisms, a more favorable barrier of 8.8 kcal mol^−1^ was observed (see Table 4 and Figure 37). The stabilization of the transition state was explained by the stabilizing hydrogen bond interactions between the H atoms of the O atom in bridge position and the departing O atom of the pNPP species (see Figure 37). In addition, the Gibbs free energies of the products were negative and therefore the reaction was exergonic, while for the non-catalyzed process the energies were positive and endergonic for the solvent-assisted mechanism or thermoneutral for the substrate-assisted mechanism. Therefore, we can state that the isomer of {Mo_2_} where the O atoms of the bridge are protonated along with the axial O atoms produces an important catalytic effect on the hydrolysis of the phosphoester bond.

The Gibbs free energy changes for the proposed steps of the overall proposed process are summarized as follows:(A)8 H_3_O^+^ + [Mo_7_O_24_]^6−^→2 [Mo_2_O_8_H_4_] + [Mo_3_O_8_]^2+^ + 8 H_2_O: ΔG° = −81.3 kcal mol^−1^(B)2 [Mo_2_O_8_H_4_] + [Mo_3_O_8_]^2+^ + 8 H_2_O→[Mo_2_O_8_H_4_] + Mo_5_O_15_ + 7 H_2_O + 2 H_3_O^+^: ΔG° = −79.3 kcal mol^−1^(C)8 H_3_O^+^ + [Mo_7_O_24_]^6−^→[Mo_2_O_8_H_4_] + Mo_5_O_15_ + 7 H_2_O + 2 H_3_O^+^: ΔG° = −160.6 kcal mol^−1^

The exergonicity of these processes leading to the in situ formation of the [Mo_2_O_8_H_4_] species along with the barrier found in Figure 37 confirmed our hypothesis concerning {Mo_2_} to be the promoter and active species for the catalysis of the phosphoester bond of the pNPP model system.

Other computational studies in our team on the catalysis of the hydrolysis of the phosphoester bond by means of Mo-oxo species focused on trying to give some explanation to the experiments of Abrantes et al. [34,35] in which the MoO_2_Cl_2_(DMF)_2_ system activated the phosphoester bond. Moreover, the polymerization of the Mo-oxo species and formation of POMs seemingly increased the rate of conversion vs. the mononuclear species. Our computations yielded some insight into different key aspects

The first step in our work was an in silico speciation study in which we elucidated the species that might exist in the medium when the promotor of the hydrolysis process was dissolved in water and tried to reproduce the experimental evidence. Having a look at the relative stability of the different mononuclear species, we could reduce the number of possibilities in the species that could have some role in the process of the promoted hydrolysis of pNPP. Therefore, the following reactions with different coordination numbers were taken into account:-Release of DMF:Mo_2_Cl_2_(DMF)_2_→MoO_2_Cl_2_(DMF)_0–1_(H_2_O)_n_-Hydrolysis of the Mo-Cl bond:MoO_2_Cl_2_(H_2_O)_0__–__2_→MoO_2_Cl_0__–__1_(OH)_1__–__2_(H_2_O)_n_-Protonation of the complex in acidic media:MoO_2_(OH)_2_(H_2_O)_0__–__2_-Intramolecular proton transfers:[MoO_2_(OH)_0–2_(H_2_O)_n_]_0–2_^+^→[MoO_x_(OH)_y_(H_2_O)_z_]_0–2_^+^

The general scheme of such rich processes in solution can be found in our previous work in reference [45]. Here, we proposed the most representative structures in Figure 38. First, we observed that the release of both DMF ligands was a favourable process, the tetrahedral coordination of Mo_2_Cl_2_ being more stable. Such processes were expected since free DMF was found in solution in the experiments of Abrantes et al. [34]. The subsequent hydrolysis of the Mo-Cl bonds was also observed experimentally, and our computations were consistent with this finding. Actually, the hydrolysis of the two Mo-Cl bonds was energetically very favourable leading to the MoO_2_(OH)_2_ complex and an acidic medium because of the consumption of OH^−^ species.

In such acidic conditions, the most stable mononuclear structure was found to be the [MoO_2_(OH)(H_2_O)_3_]^+^ species. On the other hand, we found that the tetrahedral coordination was preferred for neutral pH, whereas the octahedral coordination became more accessible when the pH was lower and the ligands were protonated. Moreover, we also observed that intramolecular proton transfers led to species that had very close energy.

Having screened the different mononuclear species present in solution, we started to study a key point of the reaction (namely, that the addition of the substrate to the promoter of the species most likely to bind the phosphate). It was pointed out in previous works [165] that the formation of [Mo_2_O_7_]^2−^ was more favourable than the phosphate addition to the mononuclear systems. The later incorporation of the phosphate to the dinuclear systems was favoured over the formation of the trinuclear structure. That ultimately lead to the formation of the Keggin anion or, in the absence of phosphate, the Lindqvist anion. Although the protonation states were not exactly the same as for our system, the molybdate condensation at different protonation states were similar and proton transfers between each site were very fast and frequent. Because the [Mo_2_O_6_(μ-O)]^2−^ resulted from the condensation of two [MoO_3_(OH)]^−^ into this dinuclear species releasing a H_2_O molecule, we studied a similar process starting from the neutral species [MoO_2_(OH)_2_] to give [Mo_2_O_4_(μ-O)(OH)_2_]. This process was compared to the addition of the phosphate substrate as shown in Figure 39. The favourable binding of [Mo_2_O_4_(μ-O)(OH)_2_] with the phosphate substrate (ΔG = −14.5 kcal mol^−1^) favoured the formation of the dinuclear species.

Besides this dinuclear species, we also searched for different bridging structures varying the coordination number of Mo as we observed that higher coordination numbers were available within this protonation state. Some double bridged structures as the [Mo_2_O_4_(μ-O)(OH_3_)(μ-OH)(OH_2_)_2_] were more favourable than [Mo_2_O_4_(μ-O)(OH)_2_] species. Such species could be related to [Mo_2_O_6_(μ-O)]^2−^ by the incorporation of a bridging OH and two H_2_O molecules, which increased the coordination number of the Mo atoms in an exergonic process (ΔG = −17.6 kcal mol^−1^). These double bridged systems were more stable when at least one of the bridging O atoms was protonated. Such systems had two H_2_O molecules that could be easily replaced by the O atoms of the pNPP (ΔG = −13.5 kcal mol^−1^) to give RDD1. This system was more stable than the direct coordination of the pNPP moiety to DM1 to give the singly bridged species RDM1, which was 11.6 kcal mol^−1^ over RDD1. Considering all these data, the main candidate for promoter of this reaction was the [Mo_2_O_4_(μ-O)(OH)_3_(μ-OH)(OH_2_)_2_] complex.

Taking into account the mechanism involving mononuclear species of Mo, the exergonic coordination (−7.6 kcal mol^−1^) of the pNPP to the metal atom of M1 by means of an O atom gave a structure with a distorted trigonal bipyramidal Mo and a tetrahedral phosphorous (RM1), as observed in Figure 39. An intramolecular proton transfer from a hydroxide of the coordination sphere of the metal to protonate the phosphate and give RM2 was favoured, releasing 6.9 kcal mol^−1^. Starting from there, we obtained structures where Mo expanded its coordination number easily. Opposite, only one structure with a phosphorane structure, having a higher energy than other intermediates. We observe in Figure 40 the Gibbs free energy profile computed for such pathway.

Once we had this structure where the two negatively charged O atoms of the phosphate were stabilized by positive charges (one proton and the metal), the formation of the phosphorane was more favourable due to the rise in electrophilicity of the phosphorous atom. For this reason, a phosphorane intermediate (IM1) was formed with low energy (18.3 kcal mol^−1^ above RM2) compared to the non-catalysed mechanism. Despite that fact, the phosphorane seemed to be barely below the transition state structures in terms of energy and its lifetime should be very short. Specifically, the formation of this hypervalent structures and the phosphoester bond breaking had a similar Gibbs free energy above the reference state (19.2 kcal mol^−1^ and 18.0 kcal mol^−1^ for TSM1 and TSM2, respectively) with the phosphorane intermediate within a narrow range of Gibbs free energy between these transition states. The cleavage of the P-O bond came along with a proton transfer from one of the protonated O atoms of the phosphate and the de-coordination of the shared hydroxide from Mo, obtaining (HPO_4_)MoO_3_ in which both centres were tetrahedral. Such pathway was more favourable than the direct pNPP hydrolysis as the interaction of the molybdate structures with the substrate was exergonic and the transition state had a lower barrier (~19 kcal mol^−1^ less than the non-catalysed process). With this information, we could explain the catalytic activity of the promoter when the concentration of promoter was very low, which disfavoured the formation of dinuclear species.

In the case of the pathway involving double bridged dinuclear species, the formation of the double bridged dinuclear species RDD1, which incorporated the phosphate, was favoured by 32.3 kcal mol^−1^ (see Figure 39). For this reason, such a pathway had a lower starting point than the mononuclear pathway and the entire pathway was favoured over the hydrolysis with respect to the mononuclear species.

The saturation of the Mo atoms coordination in RDD1 made impossible a direct interaction between them and the nucleophile. For this reason, an incoming H_2_O molecule could be stabilized by hydrogen bonds to the molybdate moiety, which favoured a correct nucleophilic attack. Eventually, this interaction could result in the abstraction of a proton by the substrate, which increased the nucleophilic character of the incoming nucleophile. This process where the substrate acted as a base catalyst is depicted in Figure 41. The proton transfer and the hydroxide attack occurred in a concerted fashion leading to the TSDD1 associative transition state localized at 22.8 kcal mol^−1^ above the reactants. The total process was exergonic by 13.9 kcal mol^−1^.

Besides this transition state, we found a dissociative transition state with a higher energy barrier of 23.4 kcal mol^−1^ that is shown in Figure 41. Such transition state was very close in energy to the associative one. On the other hand, in our other study, [44] experimental data pointed towards the existence of a dinuclear system that could be related to RDD1 through a protonation of RDD1. For this reason, we calculated the same pathway starting from RDD1prot, where both bridging O atoms were protonated. Such pathway showed a lower energy barrier and the transition state was 21.5 kcal mol^−1^ above the reactants. This transition state is shown in Figure 41 and Figure 42.

With this information we could conclude that the protonated associative mechanism (substrate assisted) with the dinuclear system was energetically favoured over the previous mononuclear mechanism. In spite of having a higher energy barrier, this mechanism started from a considerably much lower energy species, which resulted in transition states that were lower in energy than in the previous mechanism. In conditions where nucleation was favoured (i.e., low pH and enough complex concentration), this process could take place preferentially to the mononuclear pathway.

Summarizing our works on this topic on the promotion and catalysis of the hydrolysis of the phosphoester bond by means of Mo-oxo species, we have used several computational approaches along with some ESI-MS experiments to elucidate the mechanisms for the POM-catalysed hydrolysis of the phosphoester bond by using the DNA-model pNPP. We have shown that the catalytically active species responsible for the hydrolytic process was the dinuclear unit [Mo_2_O_4_(OH)_4_(pNPP)]^2−^ that exhibited a very low reaction barrier of 8.8 kcal mol^−1^ and a more exergonic reaction of −6.8 kcal mol^−1^ than the uncatalyzed processes in terms of Gibbs free energy. In contrast, the previous proposed mechanism with {Mo_5_} had a barrier of 44.1 kcal mol^−1^ and the final products had a Gibbs free energy of −3.6 kcal mol^−1^. Moreover, no adducts between {Mo_5_} and pNPP were detected in the ESI-MS experimental spectra which, in combination with the obtained computational results, suggests that the formation equilibrium of {Mo_7_} with smaller fragments like {Mo_2_} and their subsequent reassembly was the source of the hydrolytic activation rather than a rearrangement of the {Mo_5_} framework.

On the other hand, we have explained the relevant chemistry that took place in solution once the MoO_2_Cl_2_(DMF)_2_ compound was dissolved. That is, the release of DMF, the hydrolysis of the Mo-Cl bond, and the mononuclear species that would be predominant at different pH before nucleation. On the other hand, we proposed different pathways, according to the possibilities of nucleation, through which such reaction was promoted by the complex. Indeed, in the experiments with low quantity of initial complex the pH dropped to 6, and the concentration of Mo was not enough to form polynuclear species. In such conditions a slow catalysis occurred, in agreement with the proposed mononuclear pathway. In contrast, if the metal concentration was high and the medium very acidic, nucleation processes took place, which led to transient isopolyoxometalates and Keggin polyanions. We explored other pathways by considering dinuclear species that could be formed if conditions are favourable. After the very exergonic formation of such species, the surpass of the energetic barriers of the dinuclear pathways could be straightforward, which led to pNPP hydrolysis. It would lead ultimately to the formation of the Keggin anion. We highlighted that the Mo species reduced greatly the energy barrier of associative (substrate-assisted) transition states, which changed the overall preference of the phosphate hydrolysis to substrate assisted mechanisms over the solvent assisted ones.

Finally, one of the main conclusions in these studies on the use of POMs and smaller Mo-oxo species as artificial phosphoesterases was that either starting from the simplest Mo-oxo precursor with one Mo metal atom, or starting from the {Mo_7_} POM, both species may converge to dinuclear species {Mo_2_} where the bridge isomer in which the O atoms of the bridge are protonated seem to be the most active promotor for the catalysis of the phosphoester bond cleavage in the pNPP model molecule. Thus, we find that in the rich chemistry of Mo-oxo species in solution the simplest Mo oxide may act as catalyst but may also dimerize to produce a higher catalytic effect. On the other hand, the {Mo_7_} POM seems to decompose in an exergonic pathway to yield the highly active dinuclear {Mo_2_} Mo-oxo species.

## 6. Conclusions

This focused review provides a summary of the state-of-the-art contributions carried out by our team in the field of targeting canonical and non-canonical nucleic acid structures mainly by means of pure QM approaches, which usually require more computational resources and more reduced models compared to the MM and QM/MM approaches. We addressed the study of the interaction of phen based ligands and metal complexes and POMs with dDNA and GQ. In the case of phen derivatives and metal complexes containing these kinds of ligands, we expect a biophysical interaction with both kinds of DNA targets (canonical dDNA and non-canonical GQ). Nevertheless, the nature of such biophysical interaction may change depending not only on the substrate of DNA but also whether the phen ligand is isolated or included in a metal complex and whether the substitution in number and position of phen by different functional groups is of a different nature. On the other hand, when studying the interaction of POMs and Mo-oxo species with dDNA, we observed the chemical reactivity of these species in promoting and catalyzing the hydrolysis of the phosphoester bond. A lowering of the barrier is observed with respect to the conventional non-catalyzed hydrolysis and, therefore, these species may be considered as artificial pohsphoesterases.

For isolated phen derivatives interacting with dDNA models we found that, as a general trend, the nature of the interaction is mainly ruled by the attractive Δ*E_disp_* dispersion forces. However, such forces are not enough to balance the repulsive Δ*E_Pauli_* contribution and the attractive Δ*E_orb_* polarization term of the EDA, and especially the Δ*E_elstat_* electrostatic contribution (which is also attractive) are needed to obtain a negative Δ*E_int_* interaction energy. This trend for the EDA remains when phen is substituted in different number and position by -CH_3_ and =O groups but not for -OH and -NH_2_ groups capable of forming conventional hydrogen bonds with bps and the sugar and phosphate backbone of the dDNA, for which the Δ*E_elstat_* is comparable to Δ*E_disp_* or even more important. The position of substitution is more important than the number of substitutions in phen. In the case of methylation, it is explained through a kind of key and lock mechanism in which the -CH_3_ groups on position five and six are oriented in such a way that they can form the maximal number of stabilizing weak interactions not only with N and O atoms of the bps but also with the sugar and phosphate backbone. These results highlight the importance of taking into account the sugar and phosphate backbone even in any QM approach despite carrying out more time-consuming computations or using semi-empirical methods, since the seminal three-body models and the sandwich models could lead to misleading conclusions about the interaction of phen derivatives with dDNA. On the other hand, our results on the interaction energy trends for the methylated phen derivatives are in agreement with cytotoxic experimental studies, which attribute the highest cytotoxic effects to the phen derivatives substituted in positions five and six. We also explained the importance of solvent effects in the process of the intercalation of ketonic, methylated, alcoholic, and amino phen derivatives to dDNA by using continuous approaches. We observed that the consideration of the solvation penalty may change the trends on the stabilization of intercalated phen derivatives.

It was thought that classical π−π intercalators acted as electron-acceptors, whereas bps acted as electron-donors. This is confirmed after having a look at the frontier orbitals of three-body models and sandwich models (although it must be said that the charge transfer in classical π−π intercalators is very low). Nevertheless, when phen is substituted by functional groups that may form conventional hydrogen bonds not only with the bps in sandwich models but also with the sugar and phosphate backbone in ring models such as in the case of -OH and especially -NH_2_, we observed that the sense of the charge transfer depends on the orientation of the intercalation process. That is, from mg or via MG, which breaks the classical trend putting the role of electron acceptors to the intercalators and electron donors to the bps.

The abovementioned role of the sugar and phosphate backbone was even increased when we studied the intercalation of different isomers (Ax and Eq) of the [Mo(η^3^-C_3_H_5_)Br(CO)_2_(phen)] complex. Indeed, whereas for the intercalation of the [Mo(η^3^-C_3_H_5_)Br(CO)_2_(phen)] via MG the sugar and phosphate backbone did not produce any interaction with the [Mo(η^3^-C_3_H_5_)Br(CO)_2_(phen)], when the intercalation of this octahedral complex was produced via mg the sugar and phosphate backbone of the dDNA produced stabilizing weak interactions with the ancillary ligands of the [Mo(η^3^-C_3_H_5_)Br(CO)_2_(phen)] complex, which also highlighted the importance of the ancillary ligands in this process of intercalation of octahedral complexes with dDNA. These results were in agreement with the experimental results of the literature obtained from crystallographic data on the intercalation of octahedral metal complexes of Ru containing the phen derivative dzzp, in which the authors were surprised about the intercalation via mg. We explained such behavior for our [Mo(η^3^-C_3_H_5_)Br(CO)_2_(phen)] octahedral complex intercalating to the d(AGACGTCT)_2_ DNA octamer, in which the intercalation via mg is more stable, due to the more stabilizing weak interactions found in the QTAIM and NCI analyses. It must also be said that for this [Mo(η^3^-C_3_H_5_)Br(CO)_2_(phen)] complex the EDA also produced different results from those obtained with isolated phen ligand and that now the Δ*E_disp_* dispersion contribution is enough to balance the Δ*E_Pauli_* repulsive contribution. In addition, the Δ*E_orb_* orbital attractive contribution of the EDA along with the attractive Δ*E_elstat_* electrostatic contribution give to the interaction of [Mo(η^3^-C_3_H_5_)Br(CO)_2_(phen)] with dDNA a more negative Δ*E_int_* interaction energy compared to the Δ*E_int_* of isolated phen. On the other hand, when solvent effects contributions are taken into account, the stabilization between intercalation of [Mo(η^3^-C_3_H_5_)Br(CO)_2_(phen)] via mg or through the MG is more similar, which is also in agreement with experimental results found in the bibliography where octahedral complexes in solution may intercalate via MG.

Finally, in the case of the interaction of [Pt(en)(phen)]^2+^ and methylated derivatives interacting by means of the groove binding interaction with the d(GTCGAC)_2_ hexamer of dDNA, it must be said that the most stable systems have not only the phen derivative but also the en group and the Pt atom involved in the interaction since the metal complex is localized totally inside the groove. In this case, the NCI analysis shows different weak interactions between the H atoms of the en ligand and the sugar and phosphate backbone. Moreover, the Pt atom also shows quite strong interactions with the sugar and phosphate backbone apart from the interactions of the phen ligand derivative.

In the case of the GQ, we decided to tackle our studies by means of pure QM methods because for this non-canonical DNA structure improvements on force field parameters are still being carried out by different groups in the community. We used three main approaches for these QM methods (that is, semiempirical methods including dispersion effects, LS-DFT methods including van der Waals corrections, and near LS-CCSD(T) methods). We observed that the PM6-DH2 and PM7 semiempirical methods, including dispersion and the LS-DFT approach with SIESTA at the LMKLL/DZ2P level, are able to give some reasonable structures for the studied GQ from the geometrical point of view. However, taking as a reference the benchmark near LS-CCSD(T) computations at DLPNO-CCSD(T)/def2-SVP level for the interaction energies between some G-tetrads and an alkali cation, LS-DFT calculations at LMKLL/DZ2P level lead to the same trends and similar values as those obtained at the DLPNO-CCSD(T)/def2-SVP level, whereas the PM6-DH2 and PM7 semi-empirical calculations, even containing dispersion corrections and although maintaining the trends, gave interaction energy values that were very far from the DLPNO-CCSD(T)/def2-SVP benchmark results.

On the other hand, the most important finding on our studies about the interaction of the [Mo(η^3^-C_3_H_5_)Br(CO)_2_(phen)] octahedral metal complex with the GQ derived from the 2jwq PDB structure is that while the Eq isomer of the [Mo(η^3^-C_3_H_5_)Br(CO)_2_(phen)] prefers to interact with such GQ from outside by means of some kind of intercalation mode, the Ax isomer goes totally inside the non-canonical DNA structure to interact by means of end-staking with the GQ. It also interacts with the adenine tetrads adjacent to the GQ and produces their stabilization. For this reason, we could consider the eventual formation and stabilization of some kind of adenine quadruplex by means of octahedral metal complexes. Moreover, such a structure in which the Ax isomer of the [Mo(η^3^-C_3_H_5_)Br(CO)_2_(phen)] metal complex is interacting totally inside the non-canonical structure presents the most negative Δ*E_aq_* energy (−46.0 kcal mol^−1^) when including solvent effects after the EDA, which is 11.4 kcal mol^−1^ more stable than that for the most stabilized Eq isomer (Δ*E_aq_* = −34.6 kcal mol^−1^). This would be, as far as we know, the first time in which an octahedral complex is not only found forming an end-stacking interaction with a GQ but is also localized totally inside a non-canonical DNA structure, stabilizing at the same time tetrads of adenine which could form eventually some A-quadruplex.

Finally, in our studies on the topic about the use of Mo complexes and POMs as artificial phosphoesterases, interesting findings were also achieved. Indeed, we found that in the complicated chemistry of Mo species in solution, a dinuclear species is responsible for the important catalytic effect for the hydrolysis of the phosphoester bond by using the model molecule pNPP. Moreover, one of the important findings of our work about the chemistry in solution of Mo species is that starting from a simpler Mo oxide with only one metal of Mo, or starting from the {Mo_7_} POM including up to 7 Mo units, we may converge to the same dinuclear species [Mo_2_O_4_(μ-O)(OH)_2_]^0^, {Mo_2_}, which is responsible of the considerable reduction of the barrier for the phoshpoester hydrolysis. This barrier would be mainly produced by the repulsion of the negative phosphate environment with the eventual negative charge of the nucleophile. Our proposed [Mo_2_O_4_(μ-O)(OH)_2_]^0^ dinuclear Mo structure with null charge would explain the easy approach of this species to the negative charged phosphate environment. On the other hand, the stabilization of the transition state associated to the hydrolysis of the phosphoester bond by means of the hydrogen bond of the bridge OH of the [Mo_2_O_4_(μ-O)(OH)_2_]^0^ with the pNPP substrate would explain the decrease of the barrier of the process when comparing to any of the non-catalyzed mechanisms (either substrate-assisted or solvent-assisted), which produces the catalytic effect.

## 7. Outlook

To give some perspective on the use of QM approaches to treat the three topics addressed by our team concerning the interaction of flat ligands as phen derivatives (either isolated or included in metal complexes) and the use of POMs and other metal-oxo species with canonical and non-canonical DNA targets, we propose that future research should focus on the following concerns.

First of all, the development of hardware, software, and algorithms should converge, in future, to the use of not only LS-DFT but also near LS-CCSD(T) highly correlated QM methods to the correct description of these systems in which the sugar and phosphate backbone must be taken into account. In a seminal work from the early 2000s, the use of pure QM methods for the study of the intercalation of flat ligands to DNA was restricted to the three-body models and, subsequently, the sandwich models were incorporated. Our work shows that with the use of semi-empirical approaches, and specially with the use of LS-DFT, such models are obsolete and we have to consider at least the ring model (which includes the sugar and phosphate backbone). Nevertheless, for any dynamic treatment of the system, even these ring models may present some drawbacks due to the lack of the cooperative stabilizing effects of the rest of the bps; we must increase our system to at least tetramers, hexamers, octamers, etc. of dDNA for the correct description of the interaction of small molecules with dDNA, even at pure QM level. Its results are necessary, for instance, for the description of the interaction of the [Pt(en)(3,4,7,8-Me4phen)]^2+^ ligand with dDNA, since this metal complex may interact not only by means of non-reactive groove binding interaction, intercalation, or insertion but also by means of reactive interactions leading to formation of cross-links. On account of such reactivity, the QM treatment are necessary for this complex and other similar systems that are able to produce the four possible interactions with dDNA. On the other hand, the classical thought, based mainly on the analysis of the frontier orbitals of three-body and sandwich models on the nature of the flat intercalators as electron acceptors, where the bps (or the intercalator pocked) acted as electron donors should be reviewed. That is, functionalization of the intercalator by means of different groups (acidic, basic, aromatic, halogenated, etc.) and inclusion of the sugar and phosphate backbone in the model could change the electronic nature of the intercalator. Indeed, we observed that the orientation of the intercalation via mg or MG could change the electronic nature of the intercalator in terms of the HOMO and LUMO interactions. Another challenge for future research would be the use of the DLPNO-CCSD(T) highly correlated methods (and similar approaches for near LS-CCDT(T)) to study these kinds of systems. In this sense, it could be very useful to analyze interactions in terms of any energy decomposition analysis, since such highly correlated method already takes into account dispersion effects and makes the analysis independent of the different schemes we have in the bibliography on the treatment of dispersion for DFT approaches including Gimme’s corrections, highly parametrized functionals developed by Truhlar et al., long-range corrections, many-body dispersion methods, etc. Moreover, it must be said that the development of hardware, software, and algorithms may also allow the use of composite methods in which LS-DFT optimized geometries of thousands of atoms may be used along with highly correlated LS-CCSD(T) to allow the most accurate computations of interaction energies and other properties only available with the use of pure QM methods. Finally, even though we may achieve optimizations and pure QM/MD simulations with LS-DFT for systems with thousands of atoms, we may combine them in QM/MM and QM/MM/MD simulations to go one step beyond state-of-the-art systems and attempt to address more realistic systems with even more atoms.

In the case of targeting GQ structures by means of small molecules, the same perspective as for the interaction with the dDNA may be also taken into account concerning the use of pure QM methods by means of LS-DFT and LS-CCSD(T) approaches. We should also consider the use of better hardware, software, algorithms, the possibility to use composite methods based on LS-DFT and LS-CCSD(T), and the inclusion of LS-DFT methods in QM/MM optimizations and QM/MM/MD simulations with even more atoms. In the case of the GQ, the use of LS-DFT methods could be very useful since important work is still under development for parametrization of force fields to reproduce better non-canonical structures. Thus, the consideration of LS-DFT as an alternative at the QM level could prove to be very useful. In this sense, the use of semi-empirical methods including dispersion (another QM alternative) for these kinds of non-canonical structures could lead to misleading results concerning the energetics involved even though the geometries of the GQ are well-reproduced. It must be said that for studies concerning the interaction of small molecules with GQ, we have to take into account the points of interaction different from those for the dDNA. That is, we have to consider the interactions of the small molecule with the GQ by means of the groove, the loops, and the end-stacking with the bases, which makes the studies quite different from those for the dDNA. Our results concerning the interaction of the [Mo(η^3^-C_3_H_5_)Br(CO)_2_(phen)] metal complex by means of end-stacking with the GQ, totally localized inside the non-canonical DNA structures and stabilizing adenine tetrads, suggests that small molecules based on octahedral metal complexes could stabilize not only A-tetrads based in purine bases but also tetrads based on pyrimidine bases of DNA. Works in this direction could be interesting since it is known that non-canonical cytosine i-motifs of four strands exist by alternating two dDNA chains. However, we propose the stabilization of cytosine and thymine tetrads in the same plane.

Finally, in the case of the use of POMs and Mo-oxo species as artificial phosphoesterases for the catalysis of the phosphoester bond, we propose to explore other kinds of POMs based on metals of the group p and POMs based on noble transition metals, which in principle would be less toxic for the organism. On the other hand, trying to avoid the eventual toxicity to the organism for the medical use of POMs and metal-oxo species, we also propose functionalization by means of amino acids and peptides. At this point, it would be very useful to investigate how these changes in the composition of POMs and metal-oxo species and their functionalization with biomolecules affects the catalysis of the hydrolysis of the phosphoester bonds. Work is currently being conducted by our team in order to address these questions.

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
