# Peer review of "New Insights on the Interaction of Phenanthroline Based Ligands and Metal Complexes and Polyoxometalates with Duplex DNA and G-Quadruplexes"

_molecules, 2021, doi:10.3390/molecules26164737_

Round 1

Reviewer 1 Report

The review work from Sánchez-González et al. entitled “New Insights on the Interaction of Phenanthroline Based Ligands and Metal Complexes and Polyoxometalates with Duplex DNA and G-quadruplexes” is a compilation of the team work performed in the last years on small molecules and POMs interaction with duplex and G-quadruplex structures from a computational point of view. It covers an update of the computational methods used for analysing G-quadruplex binding to drugs, the intercalation and groove binding modelling of metal complexes into duplex models and the catalytic models afforded by POMs to cleave DNA. Overall, it is a complete work for theoretical chemists working with nucleic acid systems and can be useful to make a significant advance into the future methodologies to conduct computational studies with these systems. As a general view, the language has to be extensively edited and adapted to a more scientific writing such as correction of the verbal tense, change of the informal expressions and so on. In addition, the references must be reviewed since there are too many even for a review paper which are not fully understood why are present when reading it and some references are missed too (such as 185).

Author Response

The review work from Sánchez-González et al. entitled “New Insights on the Interaction of Phenanthroline Based Ligands and Metal Complexes and Polyoxometalates with Duplex DNA and G-quadruplexes” is a compilation of the team work performed in the last years on small molecules and POMs interaction with duplex and G-quadruplex structures from a computational point of view. It covers an update of the computational methods used for analysing G-quadruplex binding to drugs, the intercalation and groove binding modelling of metal complexes into duplex models and the catalytic models afforded by POMs to cleave DNA. Overall, it is a complete work for theoretical chemists working with nucleic acid systems and can be useful to make a significant advance into the future methodologies to conduct computational studies with these systems. As a general view, the language has to be extensively edited and adapted to a more scientific writing such as correction of the verbal tense, change of the informal expressions and so on. In addition, the references must be reviewed since there are too many even for a review paper which are not fully understood why are present when reading it and some references are missed too (such as 185).

We thank the Reviewer 1 for the positive and constructive comments in order to improve our manuscript for publication.

As pointed by the Reviewer 1, in the current version of the manuscript, the language has been extensively edited and adapted to a more scientific writing and the verbal tense has been corrected. Moreover, some informal expressions of the old manuscript have been changed to more formal experessions.

According to Reviewer 1 we have reduced significantly the number of references and we have 168 references in the current version of the manuscript.

Finally, we  have removed the reference 185.

Reviewer 2 Report

This article submitted by Gil et al. summarizes recent results of computational studies regarding the interaction of metal complexes with phenanthroline ligands with duplex DNA as well as G-quadruplexes. In the last part, DNA cleavage by Mo complexes and polyoxometalates is described. Although the results are scientifically sound, important and insightful, this reviewer sees some problems with the structure of the article (see below).

Major issues:

  • The article is a review, however, the structure of the article (e.g. main part is called “results”) and the attention to the detail does resemble a research article.
  • It would help the future readers of the review a lot if the article was not that detailed, but rather summarized the results and simply referred to the original work.
  • Also, the conclusion part is much to long.

Minor issues:

  • It is mentioned several times, e.g. in the introduction, that the work of the “last 6-7 years” is presented. It might be enough to mention this once, and then just go for “last couple of years” or similar.
  • Figure 1 and page 10/11: It is not clear what is meant by axial and equatorial (i.e. with respect to which ligand)?

Author Response

This article submitted by Gil et al. summarizes recent results of computational studies regarding the interaction of metal complexes with phenanthroline ligands with duplex DNA as well as G-quadruplexes. In the last part, DNA cleavage by Mo complexes and polyoxometalates is described. Although the results are scientifically sound, important and insightful, this reviewer sees some problems with the structure of the article (see below).

We thank the Reviewer 2 for his/her positive and constructive general comments on our review and the suggestions he/she proposes to improve our work.

Major issues:

The article is a review, however, the structure of the article (e.g. main part is called “results”) and the attention to the detail does resemble a research article.

It would help the future readers of the review a lot if the article was not that detailed, but rather summarized the results and simply referred to the original work.

We agree with the Reviewer 2 and in the current version of the manuscript we changed the sections and the structure of the manuscript in such a way that the document has a review structure. Moreover, we have removed some of the paragraphs of the manuscript that were too much detailed in order to address this point suggested by the Reviewer 2.

Minor issues:

It is mentioned several times, e.g. in the introduction, that the work of the “last 6-7 years” is presented. It might be enough to mention this once, and then just go for “last couple of years” or similar.

We have corrected this issue in the current version of the manuscript. Now "last 6-7 years" only appears once in the beginning of the manuscript, whereas we have removed or changed this expression to "last years" for the rest of the manuscript.

Also, the conclusion part is much to long.

In the previous version of the manuscript the conclusion part included not only the conclusions of our work but also some perspective for future studies and for this reason the conclusion part was too much long. In the current version we split this part into conclusions, which are shorter in the current version, and a new part called outlook where we put the perspectives.

Figure 1 and page 10/11: It is not clear what is meant by axial and equatorial (i.e. with respect to which ligand)?

The Reviewer 2 is right. The reference axial ligand in this pseudo-octahedral complex is the allyl. One N of phen is trans to the allyl (the other axial position) in the axial isomer. We have removed Figure 1 and we have modified the sentence to include the definition of the isomers:

Page 10, line 302: "We carried out for the first time Linear-Scaling DFT (LS-DFT) computations in an octamer of dDNA and two isomers of the [Mo(η3-C3H5)Br(CO)2phen] complex, axial (Ax, with phen nitrogen atoms trans to one CO and the allyl, top of Figure 7) and equatorial (Eq, with phen trans to both CO ligands, bottom of Figure 7)."

Reviewer 3 Report

Manuscript focuses on interaction of phenanthroline based ligands and their metal complexes with different secondary structures adopted by DNA and G-quadruplexes in particular. This work is mostly a summary of contributions carried out by the authors in the field of targeting canonical and non-canonical nucleic acid structures by QM approaches. Manuscript is really long and extensive with many citations (almost 270 in a single paper). In this way, it may be better suited for a book chapter.

Nevertheless, manuscript has its merits and deserves to be published pending decision of editors on the broad scope and its length. Authors are advised to consider points listed below.

Several of the calculated models are based on nucleosides with a proton attached to N9 or N1. Authors are probably aware of tautomerism of nucleobases that may affect stacking and other properties, including orbital energies and non-bonding interactions. Have stringent controls been performed?

G-tetrad structure as presented in Figure 7 cannot, or more appropriately, has not been demonstrated experimentally.

Sentences with self-promoting meaning such as on p.3 lines 127-142 are unnecessary and only make manuscript longer, while not contributing to the tittle subject.

Similarly, is Figure 1 with its title ‘Main topics of research addressed by our team’ promotion of the group’s research or graphical support to the research topic presented and discussed?

Graphics in Figure 4 need to be re-checked. From the current view it looks that phen ligand stacks on GC base pair rather than interacts (in coplanar geometry) via the minor groove.   

Authors are asked to consider the rest of their manuscript along these lines.

Discussion of para-nitrophenilphosphate and its hydrolysis needs justification in relation to DNA. This could be a subject of a separate manuscript.

Author Response

Manuscript focuses on interaction of phenanthroline based ligands and their metal complexes with different secondary structures adopted by DNA and G-quadruplexes in particular. This work is mostly a summary of contributions carried out by the authors in the field of targeting canonical and non-canonical nucleic acid structures by QM approaches. Manuscript is really long and extensive with many citations (almost 270 in a single paper). In this way, it may be better suited for a book chapter.

Nevertheless, manuscript has its merits and deserves to be published pending decision of editors on the broad scope and its length. Authors are advised to consider points listed below.

We thank the Reviewer 3 for the positive and constructive comments on our work.

In the new version of the manuscript we have removed some paragraphs and shortened the final version of the manuscript. Moreover, we have reduced significantly the number of references and we have 168 references in the current version of the manuscript.

Several of the calculated models are based on nucleosides with a proton attached to N9 or N1. Authors are probably aware of tautomerism of nucleobases that may affect stacking and other properties, including orbital energies and non-bonding interactions. Have stringent controls been performed?

As commented by Reviewer 3 we are aware of the tautomerism of nucleobases and its influence in stacking and other properties, including orbital energies and non-bonding interactions. Nevertheless, as also commented by Reviewer 3 our manuscript was really long and extensive in the previous version and the consideration of such tautomerism in the present manuscript could increase even more the length of the manuscript and the studies on the tautomerism were not within the objectives of this work. The original idea of this work was a review focused on the studies carried out in our team during the last years and although we know the implications of the tautomerism of the nucleobases in different properties we did not take into account, as many other authors in the bibliography, such tautomerism in our computational studies during these last years. Nevertheless, we may consider it in other forthcoming works since the present work is already long and extensive.

G-tetrad structure as presented in Figure 7 cannot, or more appropriately, has not been demonstrated experimentally.

In the original draft, Figure 7 corresponded to "phen ligand and labeled atoms for possible substitutions" and for this reason we assume that this Reviewer 3 refers not to Figure 7 but to Figure 27 in the original manuscript "G-tetrad structure including an alkali cation in the center" in his comment. It must be said that this Figure 27 (now Figure 26 in the current version of the manuscript) is just to show to the reader the main building-blocks (G-tetrads) for the formation of G-quadruplexes. Please, see the sentence in the text of page 32: "These GQ may be defined as non-canonical DNA structures where four guanine bases form a square planar array or G-tetrad (see Figure 26)." Thus, we agree with the referee that such structure, without sugar and phosphate backbone, has not been demonstrated experimentally but such structure is very useful to see which are the building-blocks (G-tetrads) to construct the experimentally demonstrated G-quadruplexes.  

Sentences with self-promoting meaning such as on p.3 lines 127-142 are unnecessary and only make manuscript longer, while not contributing to the tittle subject.

We have removed the text corresponding to p.3 lines 127-142.

Similarly, is Figure 1 with its title ‘Main topics of research addressed by our team’ promotion of the group’s research or graphical support to the research topic presented and discussed?

We have removed Figure 1 in the current version of the manuscript.

Graphics in Figure 4 need to be re-checked. From the current view it looks that phen ligand stacks on GC base pair rather than interacts (in coplanar geometry) via the minor groove.  

We agree with the Reviewer 3 and we thank for the recommandation. In the new version of the manuscript Figure 4 (now Figure 3 in the current version of the manuscript) has been re-done and current Figure 3 clearly shows the stacking of the phen ligand with the Guanine-Cytosine base pair.

Authors are asked to consider the rest of their manuscript along these lines.

Discussion of para-nitrophenilphosphate and its hydrolysis needs justification in relation to DNA. This could be a subject of a separate manuscript.

We justify the discussion of para-nitrophenylphosphate (pNPP) and its hydrolysis in relation to DNA because pNPP is a well-known model substrate to study the hydrolysis of the phosphoester bond contained in DNA and RNA backbones. Examples of the use of pNPP as model system to study the DNA and RNA phosphoester bond hydrolysis are found in the following references within the manuscript:

30. Cartuyvels, E.; Absillis, G.; Parac-Vogt, T. N. Questioning the paradigm of metal complex promoted phosphodiester hydrolysis: [Mo7O24]6- polyoxometalate cluster as an unlikely catalyst for the hydrolysis of a DNA model substrate. Commun. 2008, 85-87.

31. Van Lokeren, L.; Cartuyvels, E.; Absillis, G.; Willem, R.; Parac-Vogt, T. N. Phosphoesterase activity of polyoxomolybdates: diffusion ordered NMR spectroscopy as a tool for obtaining insights into the reactivity of polyoxometalate clusters. Commun. 2008, 2774-2776.

32. Absillis, G.; Cartuyvels, E.; Van Deun, R.; Parac-Vogt, T. N. Hydrolytic Cleavage of an RNA-Model Phosphodiester Catalyzed by a Highly Negatively Charged Polyoxomolybdate [Mo7O24]6- J. Am. Chem. Soc. 2008, 130(51), 17400-17408.

33. Absillis, G.; Van Deun, R.; Parac-Vogt, T. N. Polyoxomolybdate Promoted Hydrolysis of a DNA-Model Phosphoester Studied by NMR and EXAFS Spectroscopy. Chem. 2011, 50(22), 11552-11560.

Since the idea for the manuscript was to write a focused review on our team and we are working on 3 lines of research related to DNA:

  1. Intercalation vs. groove binding of phen derivatives.
  2. Interaction of phen derivatives with G-quadruplexes.
  3. Use of Mo-oxo species and polyoxometalates as artificial phosphoesterases.

We think that even though the third line of research includes reactivity and it is quite different from the first and second lines of research addressed in our work it is indeed related to DNA and has to be included in this review focused on the work developed during the last years in our team.

Round 2

Reviewer 2 Report

The authors have significantly improved the review article. It is now fine to be published.